# DROP: Conservative Model-based Optimization for Offline Reinforcement Learning

## Abstract

In this work, we decouple the iterative (bi-level) offline RL from the offline training phase, forming a non-iterative bi-level paradigm that avoids the iterative error propagation over two levels. Specifically, this non-iterative paradigm allows us to conduct inner-level optimization in training (*i.e.*, employing policy/value regularization), while performing outer-level optimization in testing (*i.e.*, conducting policy inference). Naturally, such paradigm raises three core questions (that are *not* fully answered by prior non-iterative offline RL counterparts like reward-conditioned policy): (Q1) What information should we transfer from the inner-level to the outer-level? (Q2) What should we pay attention to when exploiting the transferred information for the outer-level optimization? (Q3) What are the benefits of concurrently conducting outer-level optimization during testing? Motivated by model-based optimization (MBO), we proposed DROP, which fully answers the above three questions. Particularly, in the inner-level, DROP decomposes offline data into multiple subsets, and learns a MBO score model (A1). To keep safe exploitation to the score model in the outer-level, we explicitly learn a behavior embedding and introduce a conservative regularization (A2). During testing, we show that DROP permits deployment adaptation, enabling an adaptive inference across states (A3). Empirically, we evaluate DROP on various tasks, showing that DROP gains comparable or better performance compared to prior methods.

## 1 Introduction

Offline reinforcement learning (RL) (Lange et al., 2012; Levine et al., 2020) describes a task of learning a policy from previously collected static data. Due to the overestimation of values at out-of-distribution (OOD) state-actions, recent iterative offline RL methods introduce various policy/value regularization to avoid deviating from the offline data distribution (or support) in the training phase. Then, these methods directly deploy the learned policy in an online environment to test the performance. To unfold our following analysis, we term this kind of learning procedure as *iterative bi-level offline RL* (Figure 1 left), wherein the inner-level optimization refers to trying to eliminate the OOD issue by constraining the policy/value function, the outer-level optimization refers to trying to learn a better policy that will be employed at testing. Here, we use the "iterative" term to emphasize that the inner-level and outer-level are iteratively optimized in the training phase. However, without enough inner-level optimization (OOD regularization), there is still a distribution shift between the behavior policy and the policy to be evaluated. Further, due to the iterative error exploitation and propagation (Brandfonbrener et al., 2021) over the two levels, performing such an iterative bi-level optimization completely in training often struggles to learn a stable policy/value function.

In this work, we thus advocate for *non-iterative bi-level optimization* (Figure 1 right) that decouples the bi-level optimization from the training phase, namely, performing inner-level optimization (eliminating OOD) in training and performing outer-level optimization (updating policy) in testing. Intuitively, incorporating the outer-level optimization into the testing phase can eliminate the iterative error propagation over the two levels. Then, *three core questions*[1] are: (Q1) What information ("❓") should we transfer from the inner-level to the outer-level? (Q2) What should we pay special attention to when we exploit "❓" for outer-level optimization? (Q3) Notice that the outer-level optimization and the online rollout test form a new loop ("🔄"), what new benefit does this give us?

---

[1] Next we will use A1, A2, and A3 to denote our answers to the raised questions (Q1, Q2, and Q3) respectively.

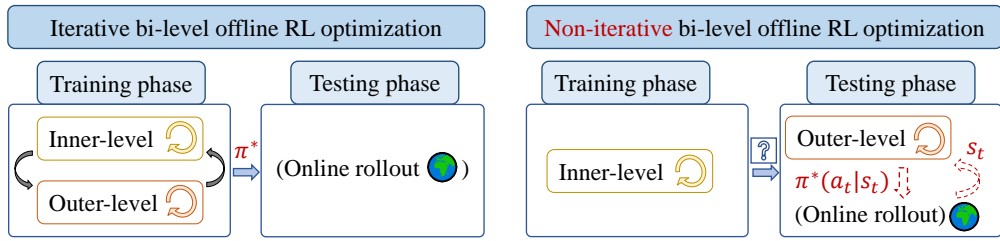

Figure 1: A framework for bi-level offline RL optimization, where the inner-level optimization refers to regularizing the policy/value function (for OOD issues) and the outer-level refers to updating the policy (for reward maximizing). Non-iterative offline RL decouples the joint optimization (of two levels) from the training phase, where "❓" transferred from the inner-level to the outer-level depends on the specific choice of algorithm used. In Table 1, we will summarize different choices for "❓".

Intriguingly, prior works under such a non-iterative framework have proposed to transfer (as "❓" in Q1) filtered trajectories (Chen et al., 2021), a reward-conditioned policy (Emmons et al., 2021; Kumar et al., 2019b), and the Q-value estimation of the behavior policy (Brandfonbrener et al., 2021; Gulcehre et al., 2021), all of which, however, partially address the aforementioned questions (we will elaborate these works in Table 1). In this work, we propose a new alternative method that transfers an embedding-conditioned (Q-value) score model and we will show that this method sufficiently answers the above questions and benefits most from the non-iterative framework.

Before introducing our method, we introduce a conceptually similar task (to the non-iterative bi-level optimization) — offline model-based optimization (MBO, Trabucco et al. (2021))[2], which aims to discover, from static input-score pairs, a new design input that will lead to the highest score. Typically, offline MBO first learns a score model that maps the input to its score via supervised regression (corresponding to inner-level optimization), and then performs inference with the learned score model (as "❓"), for instance, by optimizing the input against the learned score model via gradient ascent (corresponding to the outer-level). To enable this MBO implementation in offline RL, we are required to decompose an offline RL task into multiple sub-tasks, each of which thus corresponds to a behavior policy-return (parameters-return) pair. However, there are practical optimization difficulties when learning the score model (inner-level) and performing inference (outer-level) on high-dimensional policy's parameter space (as input for the score model). At inference, directly extrapolating the learned score model ("❓") also tends to drive the high-dimensional candidate policy (parameters) towards out-of-distribution, invalid, and low-scoring parameters (Kumar & Levine, 2020), as these are falsely and over-optimistically scored by the learned score model.

To tackle these problems, we suggest (A1) learning low-dimensional embeddings for these sub-tasks decomposed in the MBO implementation, over which we estimate an embedding-conditioned Q-value as the MBO score model ("❓" in Q1), and (A2) introduce a conservative regularization, which pushes down the predicted scores on OOD embeddings, so as to avoid over-optimistic exploitation and protect against producing unconfident embeddings when conducting outer-level optimization (policy/embedding inference). Meanwhile, (A3) learning embedding permits deployment adaptation, which means we can dynamically adjust inferred embeddings across different states in testing (aka test-time adaptation). We name our method DROP (Design fROm Policies). Compared with standard offline MBO for parameter design (Trabucco et al., 2021), deployment adaptation in DROP leverages the MDP structure of RL tasks, rather than simply conducting inference at the beginning of test rollout. Empirically, we demonstrate that DROP can effectively extrapolate a better policy that benefits from the non-iterative framework by answering the above three questions, and can achieve comparable or better performance compared to many prior offline RL algorithms.

## 2 PRELIMINARIES

### 2.1 REINFORCEMENT LEARNING AND OFFLINE REINFORCEMENT LEARNING

We model the interaction between agent and environment as a Markov Decision Process (MDP) (Sutton & Barto, 2018), denoted by the tuple $(\mathcal{S}, \mathcal{A}, R, P, \mu)$, where $\mathcal{S}$ is the state space,

---

[2]Please note that this MBO is different from the regular *model-based RL* (MBRL for short), where the *model* in MBO denotes a score model while that in MBRL deontes the transition dynamics (or reward) model.

$\mathcal{A}$ is the action space, $P : \mathcal{S} \times \mathcal{A} \times \mathcal{S} \to [0,1]$ is the transition kernel, $R : \mathcal{S} \times \mathcal{A} \to \mathbb{R}$ is the reward function, and $p_0 : \mathcal{S} \to [0,1]$ is the initial state distribution. Let $\pi \in \Pi := \{\pi : \mathcal{S} \times \mathcal{A} \to [0,1]\}$ denotes a policy. In RL, we aim to find a stationary policy that maximizes the expected discounted return $J(\pi) := \mathbb{E}_{\tau \sim \pi} [\sum_{t=0}^{\infty} \gamma^t R(\mathbf{s}_t, \mathbf{a}_t)]$ in the environment, where $\tau = (\mathbf{s}_0, \mathbf{a}_0, r_0, \mathbf{s}_1, \mathbf{a}_1, \dots )$, $r_t = R(\mathbf{s}_t, \mathbf{a}_t)$, is a sample trajectory and $\gamma \in (0,1)$ is the discount factor. We also define the state-action value function $Q^\pi(\mathbf{s}, \mathbf{a}) := \mathbb{E}_{\tau \sim \pi} [\sum_{t=0}^{\infty} \gamma^t R(\mathbf{s}_t, \mathbf{a}_t) | \mathbf{s}_0 = \mathbf{s}, \mathbf{a}_0 = \mathbf{a}]$, which describes the expected discounted return starting from state $\mathbf{s}$ and action $\mathbf{a}$ and following $\pi$ afterwards, and the state value function $V^\pi(\mathbf{s}) = \mathbb{E}_{\mathbf{a} \sim \pi(\mathbf{a}|\mathbf{s})} [Q^\pi(\mathbf{s}, \mathbf{a})]$. To maximize $J(\pi)$, actor-critic algorithm alternates between policy evaluation and improvement. Specifically, given initial $Q^0$ and $\pi^0$, it iterates

$$Q^{k+1}(\mathbf{s}, \mathbf{a}) \leftarrow \arg\min_Q \; \mathbb{E}_{(\mathbf{s},\mathbf{a},\mathbf{s}') \sim \mathcal{D}^+} \left[ \left( R(\mathbf{s}, \mathbf{a}) + \gamma \mathbb{E}_{\mathbf{a}' \sim \pi^k(\mathbf{a}'|\mathbf{s}')} \left[ Q^k(\mathbf{s}', \mathbf{a}') \right] - Q(\mathbf{s}, \mathbf{a}) \right)^2 \right], \quad (1)$$

$$\pi^{k+1}(\mathbf{a}|\mathbf{s}) \leftarrow \arg\max_\pi \; \mathbb{E}_{\mathbf{s} \sim \mathcal{D}^+, \mathbf{a} \sim \pi(\mathbf{a}|\mathbf{s})} \left[ Q^{k+1}(\mathbf{s}, \mathbf{a}) \right], \quad (2)$$

where the value function (critic) $Q(\mathbf{s}, \mathbf{a})$ is updated by minimizing the mean squared Bellman error with an experience replay dataset $\mathcal{D}^+$ and, following the deterministic policy gradient theorem (Silver et al., 2014), the policy (actor) $\pi(\mathbf{a}|\mathbf{s})$ is updated to maximize the estimated $Q^{k+1}(\mathbf{s}, \pi(\mathbf{a}|\mathbf{s}))$.

In offline RL (Levine et al., 2020), the agent is provided with a static data $\mathcal{D} = \{\tau\}$ which consists of trajectories collected by running some data-generating policies. Note that here we denote static offline data $\mathcal{D}$, distinguishing from the experience replay $\mathcal{D}^+$ in online setting. Unlike the online RL problem, where the experience $\mathcal{D}^+$ in Equation 1 can be dynamically updated, the agent, in offline RL, is not allowed to interact with the environment to collect new experience data. As a result, naively performing policy improvement as in Equation 2 may evaluate the estimated $Q^k(\mathbf{s}', \mathbf{a}')$ on actions that lie far outside of the static offline data $\mathcal{D}$, resulting in pathological values $Q^{k+1}(\mathbf{s}, \mathbf{a})$ that incur large error. Further, iterating policy evaluation and improvement will cause the learned policy $\pi^{k+1}(\mathbf{a}|\mathbf{s})$ to be biased towards out-of-distribution actions with erroneously overestimated values.

## 2.2 OFFLINE MODEL-BASED OPTIMIZATION

Model-based optimization (MBO) (Trabucco et al., 2022) aims to find an optimal design input $\mathbf{x}^*$ with a given score function $f^* : \mathcal{X} \to \mathcal{Y} \subset \mathbb{R}$, i.e., $\mathbf{x}^* = \arg\max_\mathbf{x} f^*(\mathbf{x})$. Typically, we can repeatedly query the oracle score model $f^*$ for new candidate design, until it produces the best design. However, we often do not have the oracle score function $f^*$, but are provided with a static offline dataset $\{(\mathbf{x}, y)\}$ of labeled input-score pairs. To track such offline MBO question, we can fit a parametric model $f$ to the oracle score function $f^*$ via the empirical risk minimization (ERM), $f \leftarrow \arg\min_f \mathbb{E}_{\mathbf{x}, y} \left[ (f(\mathbf{x}) - y)^2 \right]$. Then, starting from the best point in the dataset, we can perform gradient ascent on the design input and set the learned optimal design $\mathbf{x}^* = \mathbf{x}_K^\circ := \text{GradAscent}_f(\mathbf{x}_0^\circ, K)$ (for simplicity, next we will omit subscript $f$ in $\text{GradAscent}_f$), where

$$\mathbf{x}_{k+1}^\circ \leftarrow \mathbf{x}_k^\circ + \eta \nabla_\mathbf{x} f(\mathbf{x})|_{\mathbf{x}=\mathbf{x}_k^\circ}, \text{ for } k = 0, 1 \dots, K-1. \quad (3)$$

Since the aim is to find a better design input beyond all the designs in the dataset and while directly optimizing score model $f$ with ERM can not ensure new candidates (out-of-distribution design inputs) receive correct scores, thus one crucial requirement is to conduct confident extrapolation.

## 3 DROP: DESIGN FROM POLICIES

We present our framework in Figure 2. In Sections 3.1 and 3.2, we will answer questions Q1 and Q2, setting a learned MBO score model as "🖫" (A1) and introducing a conservative regularization over the score model (A2). In Section 3.3, we will answer Q3, where we show that we can conduct outer-level optimization during testing, enabling an adaptive embedding inference across states (A3).

## 3.1 TASK DECOMPOSITION

Our core idea is to explore MBO in the non-iterative bi-level offline RL framework (Figure 1 right), while capturing the structural characteristics of RL tasks and answering the raised questions (Q1, Q2, and Q3) in the introduction section. To begin with, we first decompose the offline data $\mathcal{D}$ into $N$ offline subsets $\mathcal{D}_{[N]} := \{\mathcal{D}_1, \dots, \mathcal{D}_N\}$. In other words, we decompose an offline task, learning

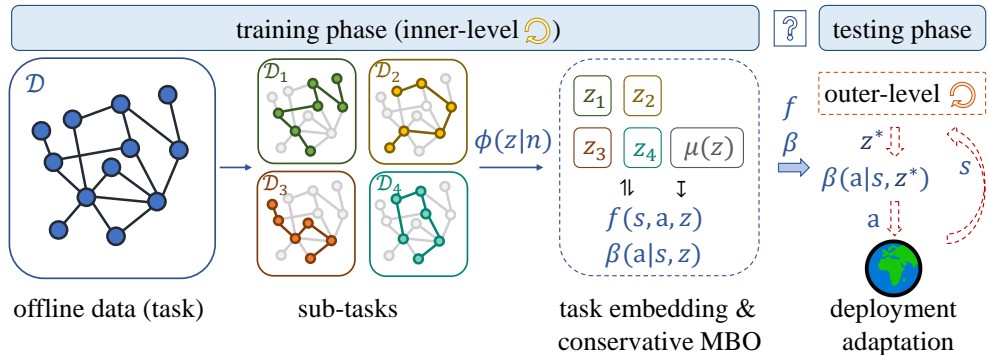

Figure 2: Overview of DROP. Given static offline dataset $\mathcal{D}$, we decompose the data into $N (= 4$ in diagram) subsets $\{\mathcal{D}_n | n = 1, \ldots, N\}$, over which we learn a task embedding $\phi(\mathbf{z}|n)$ and conduct MBO by learning multiple behavior policies (modeled by a contextual policy) $\beta(\mathbf{a}|\mathbf{s}, \mathbf{z})$ and a score model $f(\mathbf{s}, \mathbf{a}, \mathbf{z})$. During testing, at state $\mathbf{s}$, we can adapt the optimal policy (contextual variable/embedding) with $\pi^*(\mathbf{a}|\mathbf{s}) = \beta(\mathbf{a}|\mathbf{s}, \mathbf{z}^*(\mathbf{s}))$, where $\mathbf{z}^*(\mathbf{s}) = \arg\max_\mathbf{z} f(\mathbf{s}, \beta(\mathbf{a}|\mathbf{s}, \mathbf{z}), \mathbf{z})$.

with $\mathcal{D}$, into multiple offline sub-tasks, learning with $\mathcal{D}_n \in \mathcal{D}_{[N]}$ respectively. Then, for sub-task $n \in [1, N]$, we can perform behavior cloning (BC) to fit a parametric behavior policy $\beta_n : \mathcal{S} \times \mathcal{A} \to [0, 1]$ to model the corresponding subset $\mathcal{D}_n$:

$$\beta_n(\mathbf{a}|\mathbf{s}) \leftarrow \arg\max_{\beta_n} \mathbb{E}_{(\mathbf{s},\mathbf{a}) \sim \mathcal{D}_n} \big[ \log \beta_n(\mathbf{a}|\mathbf{s}) \big], \quad n = 1, \cdots, N. \tag{4}$$

Such a decomposition also comes with an additional benefit that it provides an avenue to exploit the hybrid modes in offline data $\mathcal{D}$, because that $\mathcal{D}$ is often collected using hybrid data-generating behavior policies (Fu et al., 2020), which suggests that fitting a single behavior policy may not be optimal to model the multiple modes of the offline data distribution (see Appendix C.1 for the empirical evidence). Thus, to encourage the emergence of diverse sub-tasks which capture distinct behavior modes in $\mathcal{D}$ (this is not our focus in this work[3]), we simply perform task decomposition according to the returns of trajectories in $\mathcal{D}$, heuristically ensuring that trajectories in the same sub-task share similar returns and trajectories from different sub-tasks have distinct returns.

### 3.2 Task embedding and conservative model-based optimization

**Naive model-based optimization (MBO) over behavior policies.** Benefiting from the above offline task decomposition, we can conduct MBO over a set of input-score $(\mathbf{x}, y)$ pairs, where we model the (parameters of) behavior policies $\beta_n$ as the design inputs $\mathbf{x}$ and the corresponding expected returns at initial state $J(\beta_n)$ as scores $y$. Note that, ideally, evaluating behavior policy $\beta_n$, *i.e.*, calculating $J(\beta_n) \equiv \mathbb{E}_{\mathbf{s}_0}\big[V^{\beta_n}(\mathbf{s}_0)\big]$, with subset $\mathcal{D}_n$ will never trigger the overestimation of values in the inner-level optimization. By introducing a score model $f : \Pi \to \mathbb{R}$ (as the transferred information "⍰" in Q1), we can then perform outer-level policy inference with

$$\pi^*(\mathbf{a}|\mathbf{s}) \leftarrow \arg\max_\pi f(\pi), \text{ where } f = \arg\min_f \mathbb{E}_n \left[ \big( f(\beta_n) - J(\beta_n) \big)^2 \right]. \tag{5}$$

However, directly performing optimization-based inference (outer-level optimization), $\max_\pi f(\pi)$, will quickly find an invalid input for which the learned score model $f$ outputs erroneously large values (Q2). Furthermore, it is particularly severe if we perform the inference directly over the parameters of policies, accounting for the fact that the parameters of input behavior policies lie on a narrow manifold in a high-dimensional parametric space (Kumar & Levine, 2020).

**Task embedding.** To enable feasible policy inference, we propose to decouple the MBO techniques from the high-dimensional space of policy parameters. We achieve this by learning a latent embedding space $\mathcal{Z}$ with an information bottleneck $(\dim(\mathcal{Z}) \ll \min(N, \dim(\Pi)))$, conditioned on the sub-task id $n$, from which the high-dimensional parameters of behavior policies can be inferred.

---

[3]We compare three different decomposition rules in Appendix C.2. Further, in Appendix C.4, we adopt CVAE to conduct automatic task decomposition (treating each trajectory as an individual task).

We can thus use the embedding $\mathbf{z} \in \mathcal{Z}$ to represent sub-tasks (or the corresponding behavior policies). Formally, we learn a task embedding[4] $\phi : \mathbb{R}^N \times \mathcal{Z} \to [0, 1]$ and a *contextual* behavior policy $\beta : \mathcal{S} \times \mathcal{Z} \times \mathcal{A} \to [0, 1]$, which replaces $N$ separate behavior policies in Equation 4:

$$\beta(\mathbf{a}|\mathbf{s}, \mathbf{z}), \phi(\mathbf{z}|n) \leftarrow \arg\max_{\beta, \phi} \mathbb{E}_{\mathcal{D}_n \sim \mathcal{D}_{[N]}} \mathbb{E}_{(\mathbf{s}, \mathbf{a}) \sim \mathcal{D}_n} \big[ \log \beta(\mathbf{a}|\mathbf{s}, \phi(\mathbf{z}|n)) \big]. \qquad (6)$$

**Conservative model-based optimization.** In principle, by substituting the learned task embedding $\phi(\mathbf{z}|n)$ and the contextual behavior policy $\beta(\mathbf{a}|\mathbf{s}, \mathbf{z})$ into the original objective in Equation 5, we can then conduct MBO over the embedding space: learning $f : \mathcal{Z} \to \mathbb{R}$ with $\min_f \mathbb{E}_{n, \phi(\mathbf{z}|n)} \big[ (f(\mathbf{z}) - J(\beta_n))^2 \big]$, and setting the optimal embedding $\mathbf{z}^* = \arg\max_{\mathbf{z}} f(\mathbf{z})$ and the corresponding policy $\pi^*(\mathbf{a}|\mathbf{s}) = \beta(\mathbf{a}|\mathbf{s}, \mathbf{z}^*)$, where $\mathbf{z}^*$ can be inferred with gradient ascent as in Equation 3. However, we must deliberate a new distribution shift in the $\mathcal{Z}$-space, stemming from the original distribution shift in the parametric space when directly optimizing Equation 5.

Motivated by the energy model (LeCun et al., 2006) and the conservative regularization (CQL, Kumar et al. (2020)), we introduce the conservative score model learning, additionally regularizing the scores of out-of-distribution embeddings $\mu(\mathbf{z})$:

$$f \leftarrow \arg\min_f \mathbb{E}_{n, \phi(\mathbf{z}|n)} \big[ (f(\mathbf{z}) - J(\beta_n))^2 \big], \text{ s.t. } \mathbb{E}_{\mu(\mathbf{z})} [f(\mathbf{z})] - \mathbb{E}_{n, \phi(\mathbf{z}|n)} [f(\mathbf{z})] \leq \eta. \qquad (7)$$

Intuitively, as long as the scores of out-of-distribution embeddings $\mathbb{E}_{\mu(\mathbf{z})} [f(\mathbf{z})]$ is lower than that of in-distribution embeddings $\mathbb{E}_{n, \phi(\mathbf{z}|n)} [f(\mathbf{z})]$ (up to a threshold $\eta$), conducting embedding inference with $\mathbf{z}^* = \arg\max_{\mathbf{z}} f(\mathbf{z})$ would produce the best and confident solution, avoiding towards embeddings that are far away from the training set $\{\phi(\mathbf{z}|n), n = 1, \dots, N\}$.

Now that we have reframed the non-iterative bi-level offline RL problem as one of offline MBO: in the inner-level optimization (Q1), we set the practical choice for "❓" as the learned score model $f$ (A1); in the outer-level optimization (Q2), we introduce task embedding and conservative regularization to avoid over-optimistic exploitation when exploiting $f$ for policy/embedding inference (A2). In next section, we will show how to slightly change the form of the score model $f$, so as to leverage the (MDP) structural characteristic (loop "🔁") of RL tasks and answer the left Q3.

## 3.3 DEPLOYMENT ADAPTATION

Recalling that we update $f(\mathbf{z})$ to regress the value at initial state $\mathbb{E}_{\mathbf{s}_0} \big[ V^{\beta_n}(\mathbf{s}_0) \big]$ in Equation 7, we then conduct outer-level inference with $\mathbf{z}^* = \arg\max_{\mathbf{z}} f(\mathbf{z})$ and rollout the $z^*$-conditioned policy $\pi^*(\mathbf{a}|\mathbf{s}) := \beta(\mathbf{a}|\mathbf{s}, \mathbf{z}^*)$ until the end of rollout episode at deployment (testing). In essence, this inference produces an extrapolation over the distribution of the behavior policies (corresponding to embeddings). Going beyond the (outer-level) inference only at the initial state, we propose that a implementation can benefit by performing inference at any rollout state in testing (A3).

To enable deployment adaptation, we model the score model with $f : \mathcal{S} \times \mathcal{A} \times \mathcal{Z} \to \mathbb{R}$, taking a state-action as extra input. Then, we encourage the score model to regress the values of behavior policies over all state-action pairs in each sub-task, $\min_f \mathbb{E}_{n, \phi(\mathbf{z}|n)} \mathbb{E}_{(\mathbf{s}, \mathbf{a}) \sim \mathcal{D}_n} \big[ (f(\mathbf{s}, \mathbf{a}, \mathbf{z}) - Q^{\beta_n}(\mathbf{s}, \mathbf{a}))^2 \big]$. For simplicity, instead of learning an additional value function $Q^{\beta_n}$ for each behavior policy, we learn the score model directly with the TD-error used for learning the value function $Q^{\beta_n}(\mathbf{s}, \mathbf{a})$ as in Equation 1, together with the conservative regularization in Equation 7:

$$f \leftarrow \arg\min_f \mathbb{E}_{\mathcal{D}_n \sim \mathcal{D}_{[N]}} \mathbb{E}_{(\mathbf{s}, \mathbf{a}, \mathbf{s}', \mathbf{a}') \sim \mathcal{D}_n} \big[ (R(\mathbf{s}, \mathbf{a}) + \gamma \bar{f}(\mathbf{s}', \mathbf{a}', \phi(\mathbf{z}|n)) - f(\mathbf{s}, \mathbf{a}, \phi(\mathbf{z}|n)))^2 \big], \qquad (8)$$

$$\text{s.t. } \mathbb{E}_{n, \mu(\mathbf{z})} \mathbb{E}_{\mathbf{s} \sim \mathcal{D}_n, \mathbf{a} \sim \beta(\mathbf{a}|\mathbf{s}, \mathbf{z})} [f(\mathbf{s}, \mathbf{a}, \mathbf{z})] - \mathbb{E}_{n, \phi(\mathbf{z}|n)} \mathbb{E}_{\mathbf{s} \sim \mathcal{D}_n, \mathbf{a} \sim \beta(\mathbf{a}|\mathbf{s}, \mathbf{z})} [f(\mathbf{s}, \mathbf{a}, \mathbf{z})] \leq \eta,$$

where $\bar{f}$ denotes a target network and we update the target $\bar{f}$ with soft updates: $\bar{f} = (1 - \upsilon)\bar{f} + \upsilon f$.

In testing, we thus can dynamically adapt the outer-level optimization, setting policy inference with $\pi^*(\mathbf{a}|\mathbf{s}) = \beta(\mathbf{a}|\mathbf{s}, \mathbf{z}^*(\mathbf{s}))$, where $z^*(\mathbf{s}) = \arg\max_z f(\mathbf{s}, \beta(\mathbf{a}|\mathbf{s}, \mathbf{z}), \mathbf{z})$. Specifically, at any state $\mathbf{s}$ in the deployment phase, we perform gradient ascent to find the optimal behavior embedding $z^*(\mathbf{s}) = \mathbf{z}_K^{\circ}(\mathbf{s}) := \text{GradAscent}(\mathbf{s}, \mathbf{z}_0^{\circ}, K)$, where $\mathbf{z}_0^{\circ}$ is the starting point and

$$\mathbf{z}_{k+1}^{\circ}(\mathbf{s}) \leftarrow \mathbf{z}_k^{\circ}(\mathbf{s}) + \eta \nabla_{\mathbf{z}} f(\mathbf{s}, \beta(\mathbf{a}|\mathbf{s}, \mathbf{z}), \mathbf{z}))|_{\mathbf{z} = \mathbf{z}_k^{\circ}}, \text{ for } k = 0, 1 \dots, K - 1. \qquad (9)$$

---

[4]We feed the one-hot encoding of the sub-task specification ($n = 1, \dots, N$) into the embedding network $\phi$.

Table 1: Comparison of five non-iterative bi-level offline algorithms, where $R(\cdot)$ denotes the return of sampling $\tau$ or starting from $(\mathbf{s}, \mathbf{a})$, the checkmark in A2 indicates whether the exploitation (outer-level) to "⬜" is regularized and that in A3 indicates whether deployment adaptation is supported.

| | Inner-level | Outer-level | "⬜" in A1 | A2 | A3 |
|---|---|---|---|---|---|
| F-BC | filter $\tau$ with high $R(\tau)$ | behavior cloning | filtered $\{\tau\}$ | ✗ | ✗ |
| RvS-R | $\min_\pi -\mathbb{E}\left[\log \pi(\mathbf{a}\vert\mathbf{s}, R(\tau))\right]$ | handcraft $R_{\text{target}}$ | $\pi(\mathbf{a}\vert\mathbf{s}, \cdot)$ | ✗ | ✗ |
| Onestep | $\min_Q \mathcal{L}(Q(\mathbf{s}, \mathbf{a}), R(\mathbf{s}, \mathbf{a}))$ | $\arg\max_{\mathbf{a}} Q_\beta(\mathbf{s}, \beta(\mathbf{a}\vert\mathbf{s}))$ | $Q_\beta(\mathbf{s}, \mathbf{a})$ | ✗ | ✔ |
| COMs | $\min_f \mathcal{L}(f(\beta_\tau), R(\tau))$ | $\arg\max_\beta f(\beta)$ | $f(\beta)$ | ✔ | ✗ |
| DROP | $\min_f \mathcal{L}(f(\mathbf{s}, \mathbf{a}, \mathbf{z}), R(\mathbf{s}, \mathbf{a}, \mathbf{z}))$ | $\arg\max_{\mathbf{z}} f(\mathbf{s}, \beta(\mathbf{a}\vert\mathbf{z}), \mathbf{z})$ | $f(\mathbf{s}, \mathbf{a}, \mathbf{z})$ | ✔ | ✔ |

### 3.4 CONNECTION TO PRIOR NON-ITERATIVE OFFLINE COUNTERPARTS

In Table 1, we summarize the comparison with prior representative non-iterative offline RL methods. Intuitively, our DROP (leveraging returns to decompose $\mathcal{D}$) is similar in spirit to F-BC and RvS-R (Chen et al., 2021; Emmons et al., 2021), both of which use return $R(\tau)$ to guide the inner-level optimization. However, both F-BC and RvS-R leave Q2 unanswered. In outer-level, F-BC can not enable policy extrapolation, which heavily relies on the data quality in offline tasks, and RvS-R needs to handcraft a target return (as the contextual variable for $\pi(\mathbf{a}\vert\mathbf{s}, \cdot)$), which also probably triggers the potential distribution shift between the hand-crafted contextual variable and that used for learning the contextual policy (see examples in Figure 6 of Emmons et al. (2021)).

Diving deeper into the bi-level optimization, we can also find DROP combines the advantages of Onestep (Brandfonbrener et al., 2021) and COMs (Trabucco et al., 2021), where Onestep performs outer-level optimization in action space ($\arg\max_{\mathbf{a}}$), COMs performs that in parameter space ($\arg\max_\beta$), while our DROP performs that in embedding space ($\arg\max_{\mathbf{z}}$). As a result, the choice of $f(\mathbf{s}, \mathbf{a}, \mathbf{z})$ in DROP allows us to conduct safe exploitation over "⬜" in outer-level (Q2) and leverage the structural characteristic of RL task (the loop "🔄" in Q3), rather than simply conducting outer-level optimization at initial states as in COMs (corresponding to the objective in Equation 5).

### 3.5 PRACTICAL IMPLEMENTATION

---

**Algorithm 1** DROP (Training)

**Require:** Dataset of trajectories, $\mathcal{D} = \{\tau\}$.
1: Initialize $\phi(\mathbf{z}\vert n)$, $\beta(\mathbf{a}\vert\mathbf{s}, \mathbf{z})$, and $f(\mathbf{s}, \mathbf{a}, \mathbf{z})$.
2: Decompose $\mathcal{D}$ into $N$ sub-sets $\mathcal{D}_{[N]}$.
3: **while** not converged **do**
4:     Sample a sub-task: $\mathcal{D}_n \sim \mathcal{D}_{[N]}$.
5:     Learn $\phi$, $\beta$, and $f$ with Equations 6 and 8.
6: **end while**
**Return:** $\beta(\mathbf{a}\vert\mathbf{s}, \mathbf{z})$ and $f(\mathbf{s}, \mathbf{a}, \mathbf{z})$.

---

**Algorithm 2** DROP (Testing / Deployment)

**Require:** Env, $\beta(\mathbf{a}\vert\mathbf{s}, \mathbf{z})$, and $f(\mathbf{s}, \mathbf{a}, \mathbf{z})$.
1: $\mathbf{s}_0 = $ Env.Reset().
2: **while** not done **do**
3:     Inference (deployment adaptation):
    $\mathbf{z}^*(\mathbf{s}_t) = \arg\max_{\mathbf{z}} f(\mathbf{s}_t, \beta(\mathbf{a}_t\vert\mathbf{s}_t, \mathbf{z}), \mathbf{z})$.
4:     Sample action: $\mathbf{a}_t \sim \beta(\mathbf{a}_t\vert\mathbf{s}_t, \mathbf{z}^*(\mathbf{s}_t))$.
5:     Step Env: $\mathbf{s}_{t+1} \sim P(\mathbf{s}_{t+1}\vert\mathbf{s}_t, \mathbf{a}_t)$.
6: **end while**

---

We now summarize the DROP algorithm (see Algorithm 1 for the training phase and Algorithm 2 for the testing phase). During training (inner-level optimization), we alternate between updating $\phi(\mathbf{z}\vert n)$, $\beta(\mathbf{a}\vert\mathbf{s}, \mathbf{z})$, and $f(\mathbf{s}, \mathbf{a}, \mathbf{z})$, wherein we update $\phi$ with both maximum likelihood loss and TD-error loss in Equations 6 and 8. During testing (outer-level optimization), for each state $\mathbf{s}$, we use the gradient ascent in Equation 9 to choose the optimal embedding $z^*(\mathbf{s})$. Instead of simply sampling a single starting point $\mathbf{z}_0^\circ$, we choose $N$ starting points corresponding to all the embeddings $\{\mathbf{z}_n \vert n = 1, \ldots, N\}$ of sub-tasks, and then choose the optimal $\mathbf{z}^*(\mathbf{s})$ from those *updated embeddings* for which the learned $f$ outputs the highest score: $\mathbf{z}^*(\mathbf{s}) = \arg\max_{\mathbf{z}} f(\mathbf{s}, \beta(\mathbf{a}\vert\mathbf{s}, \mathbf{z}), \mathbf{z})$ s.t. $\mathbf{z} \in \{\text{GradAscent}(\mathbf{s}, \mathbf{z}_n, K)\vert n = 1, \ldots, N\}$. Then, we sample action from $\pi^*(\mathbf{a}\vert\mathbf{s}) := \beta(\mathbf{a}\vert\mathbf{s}, \mathbf{z}^*(\mathbf{s}))$. For more optimization (training/testing) details, we refer the reader to Appendix D.

## 4 RELATED WORK

**Offline RL.** In offline RL, learning with static offline data is prone to exploiting out-of-distribution (OOD) state-action pairs and producing over-estimation of values, which makes vanilla iterative

policy learning and value optimization challenging (Rashidinejad et al., 2021). To eliminate the problem, a number of methods have been explored, in essence, by either *introducing a policy/value regularization in the iterative loop* or *trying to eliminate the iterative procedure itself*.

*Iterative methods:* Sticking with the normal iterative updates in RL, offline policy regularization methods aim to keep the learning policy to be close to the behavior policy under a probabilistic distance (Cang et al., 2021; Fujimoto & Gu, 2021; Kostrikov et al., 2021a; Kumar et al., 2019a; Liu et al., 2022; Nair et al., 2020a; Peng et al., 2019; Siegel et al., 2020; Wu et al., 2019; Zhang et al., 2021). Some works also conduct implicit policy regularization with variants of importance sampling Lee et al. (2021); Liu et al. (2019); Nachum et al. (2019). Besides regularizing policy, it is also feasible to constrain the substitute value function in the iterative loop. Methods constraining the value function aim at mitigating the over-estimation, which typically introduces pessimism to the prediction of the Q-values (Chebotar et al., 2021; Jin et al., 2021; Kumar et al., 2020; Li et al., 2022; Ma et al., 2021a;b) or penalizes the value with an uncertainty quantification (An et al., 2021; Bai et al., 2022; Rezaeifar et al., 2021; Wu et al., 2021), making the value for out-of-distribution state-actions more conservative. Similarly, another branch of model-based methods (Kidambi et al., 2020; Yu et al., 2020; 2021b; Rigter et al., 2022) also perform iterative bi-level updates, alternating between regularized evaluation and improvement. Different from these works, DROP only evaluates values of behavior policies in the inner-level, avoiding error propagation between two levels.

*Non-iterative methods:* Another complementary line of work studies how to eliminate the iterative updates, which simply casts RL as a *weighted* or *conditional* imitation learning problem (Q1). Derived from the behavior-regularization RL (Geist et al., 2019; Vieillard et al., 2020), *the former* conducts weighted behavior cloning: first learn a value function for the behavior policy, then weigh the state-action pairs with the learned values or advantages (Abdolmaleki et al., 2018; Chen et al., 2019; Wang et al., 2020; Peng et al., 2019). Besides, some works also propose implicitly behavior policy regularization that also avoids estimating the value of new candidate policies, initializing the learning policy with a behavior policy Matsushima et al. (2020) or performing only a "one-step" update (policy improvement) over the behavior policy (Fujimoto et al., 2019; Gulcehre et al., 2021). For *the latter*, this branch method typically builds upon the hindsight information matching (Andrychowicz et al., 2017; Eysenbach et al., 2020; Pong et al., 2018; Wan et al., 2021), assuming that the future trajectory information can be useful to infer the middle decision that leads to the future and thus relabeling the trajectory with the reached states or returns. Due to the simplicity and stability, RvS-based methods thus advocate for learning a goal-conditioned or reward-conditioned policy (Chen et al., 2021; Ding et al., 2019; Emmons et al., 2021; Furuta et al., 2021; Janner et al., 2021; Lin et al., 2022; Srivastava et al., 2019; Yang et al., 2022) with supervised learning. However, these works do not fully exploit the non-iterative bi-level framework and fail to answer the proposed questions, which either does not regularize the inner-level optimization before exploiting "⌗" in the outer-level (Q2), or does not support the deployment adaptation in testing (Q3).

**Offline model-based optimization (MBO).** Similar to offline RL, the main challenge of MBO is to reason about uncertainty and OOD values (Brookes et al., 2019; Fannjiang & Listgarten, 2020), since a direct gradient-ascent against the learned score model can easily produce invalid inputs that are falsely and highly scored. To counteract the effect of model exploitation, prior works introduce various techniques, including normalized maximum likelihood estimation (Fu & Levine, 2021), model inversion networks (Kumar & Levine, 2020), local smoothness prior (Yu et al., 2021a), and conservative objective models (COMs) (Trabucco et al., 2021). Compared to COMs, DROP shares similarity with the conservative model, but instantiates on the embedding space instead of the parameter space. Such difference is nontrivial, not only because DROP allows OOD sampling (aimed at pessimism) directly in embedding space, avoiding an adversarial training as in COMs, but also because DROP allows deployment adaptation, enabling dynamical inference across states in testing.

## 5 EXPERIMENTS

In this section, we present our empirical results. We first give examples to illustrate the deployment adaptation. Then we evaluate DROP against prior offline RL algorithms on D4RL benchmark. Finally, we compare DROP with prior (offline) latent-based baselines. For more ablation studies wrt to the decomposition rules and the conservative regularization, we refer readers to the appendix.

**Illustration of deployment adaptation.** To better understand the deployment adaptation of DROP, we include four comparisons that exhibit different embedding inference rules at testing:

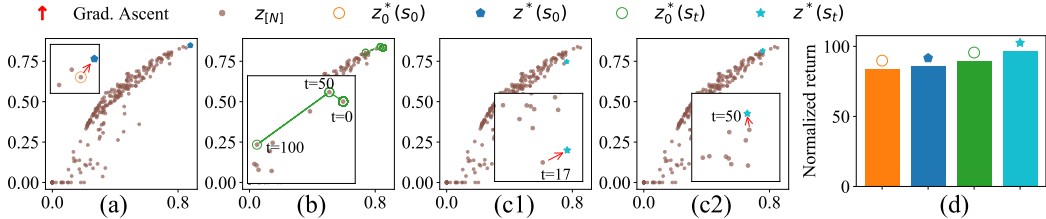

Figure 3: Visualization of the embedding inference (a, b, c1, c2) and performance comparison (d). $\mathbf{z}_{[N]}$ denotes embeddings of all behavior policies; $\mathbf{z}_0^*(\mathbf{s}_0)$, $\mathbf{z}^*(\mathbf{s}_0)$, $\mathbf{z}_0^*(\mathbf{s}_t)$ and $\mathbf{z}^*(\mathbf{s}_t)$ denote the selected embeddings in DROP-Best, DROP-Grad, DROP-Best-Ada and DROP-Grad-Ada respectively.

(1) DROP-Best: At initial state $\mathbf{s}_0$, we choose the best embedding from those embeddings of behavior policies, $\mathbf{z}_0^*(\mathbf{s}_0) = \arg\max_z f(\mathbf{s}_0, \beta(\mathbf{a}_0|\mathbf{s}_0, \mathbf{z}), \mathbf{z})$ s.t. $\mathbf{s} \in \mathbf{z}_{[N]} := \{\mathbf{z}_1, \ldots, \mathbf{z}_N\}$, and keep this embedding fixed for the entire episode, *i.e.*, setting $\pi^*(\mathbf{a}_t|\mathbf{s}_t) = \beta(\mathbf{a}_t|\mathbf{s}_t, \mathbf{z}_0^*(\mathbf{s}_0))$.

(2) DROP-Grad: At initial state $\mathbf{s}_0$, we conduct inference (gradient ascent on starting point $\mathbf{z}_0^*(\mathbf{s}_0)$) with $\mathbf{z}^*(\mathbf{s}_0) = \arg\max_z f(\mathbf{s}_0, \beta(\mathbf{a}_0|\mathbf{s}_0, \mathbf{z}), \mathbf{z})$, and keep this embedding fixed throughout the rollout.

(3) DROP-Best-Ada: We adapt the contextual policy by setting $\pi^*(\mathbf{a}_t|\mathbf{s}_t) = \beta(\mathbf{a}_t|\mathbf{s}_t, \mathbf{z}_0^*(\mathbf{s}_t))$, where we choose the best embedding $\mathbf{z}_0^*(\mathbf{s}_t)$ directly from those embeddings of behavior policies for which the score model outputs the highest score, *i.e.*, $\mathbf{z}_0^*(\mathbf{s}_t) = \arg\max_{\mathbf{z}} f(\mathbf{s}_t, \beta(\mathbf{a}_t|\mathbf{s}_t, \mathbf{z}), \mathbf{z})$ s.t. $\mathbf{z} \in \mathbf{z}_{[N]}$.

(4) DROP-Grad-Ada (gradient-based adaptation as described in Section 3.5): We set $\pi^*(\mathbf{a}_t|\mathbf{s}_t) = \beta(\mathbf{a}_t|\mathbf{s}_t, \mathbf{z}^*(\mathbf{s}_t))$ and choose the best embedding from those *updated embeddings* of behavior policies, *i.e.*, $\mathbf{z}^*(\mathbf{s}_t) = \arg\max_{\mathbf{z}} f(\mathbf{s}_t, \beta(\mathbf{a}_t|\mathbf{s}_t, \mathbf{z}), \mathbf{z})$ s.t. $\mathbf{z} \in \{\text{GradAscent}(\mathbf{s}_t, \mathbf{z}_n, K)|n = 1, \ldots, N\}$.

In Figure 3, we visualize the four different inference rules and report the corresponding performance in the halfcheetah-medium-expert task (Fu et al., 2020). In Figure 3 (a), we set the starting point as the best embedding $\mathbf{z}_0^*(\mathbf{s}_0)$ in $\mathbf{z}_{[N]}$, and perform gradient ascent to find the optimal $\mathbf{z}_0^*(\mathbf{s}_0)$ for DROP-Grad. In Figure 3 (b), we can find that at different time steps, DROP-Best-Ada chooses different embeddings (as contextual variables for $\beta(\mathbf{a}_t|\mathbf{s}_t, \cdot)$). At a high level, performing such dynamical inference enables us to combine different embeddings, *switching* behavior policies at different states. Further, in Figure 3 (c1, c2), we find that the additional inference (with gradient ascent) in DROP-Grad-Ada allows to extrapolate beyond the embeddings of behavior policies, and thus results in sequential composition of *new embeddings* (policies) across different states. For practical impacts of these different inference rules, we provide the performance comparison in Figure 3 (d), where we can find that performing gradient-based optimization (*-Grad-*) outperforms the natural selection among these embeddings of behavior policies in sub-tasks (*-Best-*), and rollout with adaptive embedding inference (DROP-*-Ada) outperforms that with fixed embeddings (DROP-*).

**Empirical performance on benchmark tasks.** We evaluate DROP on a number of tasks from the D4RL dataset and make comparison with both prior iterative and non-iterative offline algorithms[5].

Considering that DROP follows the non-iterative offline RL paradigm, we compare DROP with prior non-iterative offline baselines (BC, F-BC, DT (Chen et al., 2021), RvS-R, Onestep, and COMs) in the main paper. Compared with our DROP, these baselines do not fully answer the raised questions (see Table 1), which either does not regularize the inner-level optimization before exploiting "❓" in outer-level (Q2), or does not support the deployment adaptation in testing (Q3). Moreover, we also provide the comparison with CQL, which inspires us to design the conservation in Equation 7. In Table 2, we show the evaluation results for AntMaze-* and Gym-*-medium-* tasks in D4RL *-v2, where we can find DROP (-Grad-Ada) achieves better performance than these non-iterative offline RL baselines overall. Compared with CQL, DROP shows superior performance in AntMaze-large-* and Gym-*-medium-expert (m.-exp.) tasks, while leads inferior performance in AntMaze-medium-* and Gym-*-medium-replay (m.-rep.) tasks. As an extension, we also design DROP+CVAE implementation (see motivation in next paragraph and details in Appendix C.4), which further improves DROP's performance and retains superior/comparable performance in all tasks.

**Comparison with latent policy methods.** Note that one additional merit of DROP is that it naturally accounts for hybrid modes in $\mathcal{D}$ by conducting task decomposition in inner-level, we thus

---

[5]Due to page limit, we mainly provide the comparison with prior iterative baselines in Appendix E

Table 2: Comparison with non-iterative methods on D4RL (*-v2). For all results of our method, we average the normalized returns across 5 seeds; for each seed, we run 10 evaluation episodes. For comparison, we use ▲ and ▲ to denote DROP-Grad-Ada achieves better performance compared with Onestep and COMs (most related baselines in Table 1) respectively. (BA: Best-Ada, GA: Grad-Ada)

| | Tasks | BC | F-BC | DT | RvS-R | Onestep ▲ | COMs ▲ | CQL | DROP BA | DROP GA | DROP+CVAE GA |
|---|---|---|---|---|---|---|---|---|---|---|---|
| antmaze | umaze | 54.6 | 60 | 65.6 | 64.4 | 64.3 | 63.3 | 74 | 78 | 80▲▲ | **90.5 ± 2.4** |
| | umaze-diverse | 45.6 | 46.5 | 51.2 | 70.1 | 60.7 | 46.7 | 84 | 62 | 66▲▲ | **85.1 ± 7.8** |
| | medium-play | 0 | 42.1 | 1 | 4.5 | 0.3 | 40 | 61.2 | 34 | 30▲ | **68.2 ± 16.5** |
| | medium-diverse | 0 | 37.2 | 0.6 | 7.7 | 0 | 26.7 | 53.7 | 24 | 30▲▲ | **75.4 ± 9.4** |
| | large-play | 0 | 28 | 0 | 3.5 | 0 | 33.3 | 15.8 | 36 | 42▲▲ | **50.1 ± 13.6** |
| | large-diverse | 0 | 34.3 | 0.2 | 3.7 | 0 | 10 | 14.9 | 20 | 26▲▲ | **52.2 ± 12.0** |
| m.-rep. | walker2d | 26 | 62.5 | 66.6 | 60.6 | 66.4 | 33.9 | **86.1** | 60.9 | 61.9 ▲ | 83.5 ± 1.2 |
| | hopper | 18.1 | 75.9 | 82.7 | 73.5 | 77.3 | 49.7 | **97.8** | 83.4 | 87.4▲▲ | 96.3 ± 2.5 |
| | halfcheetah | 36.6 | 40.6 | 36.6 | 38 | 38.4 | 41.4 | 47.3 | 40.4 | 40.3▲ | **50.9 ± 1.6** |
| m.-exp. | walker2d | 107.5 | 109 | 108.1 | 106 | **111.8** | 93.5 | 109.5 | 106.8 | 106.9 ▲ | 109.3 ± 0.4 |
| | hopper | 52.5 | **110.9** | 107.6 | 101.7 | 81.4 | 109.4 | 102 | 102.5 | 105.9▲ | 107.2 ± 1.7 |
| | halfcheetah | 55.2 | 92.9 | 86.8 | 92.2 | 77 | 76 | 85.8 | 88.5 | 88.9▲▲ | **102.2 ± 1.5** |
| | total | 396.1 | 739.9 | 607.0 | 625.9 | 577.6 | 623.9 | 832.1 | 736.5 | 765.3 | **970.9** |

compare DROP to latent policy methods (PLAS (Zhou et al., 2020) and LAPO (Chen et al., 2022)) that use conditional variational autoencoder (CVAE) to model offline data and also account for multi-modes in offline data. Essentially, both our DROP and baselines (PLAS and LAPO) learn a latent policy in the inner-level optimization, except that we adopt the non-iterative bi-level learning while baselines are instantiated under the iterative paradigm. By answering Q3, DROP permits deployment adaptation, enabling us to dynamically switch/stitch "skills" (latent-policy/behaviors as shown in Figure 3) and encouraging high-level abstract exploration in testing. However, the aim of introducing the latent policy in PLAS and LAPO is to regularize the inner-level optimization, which fairly answers Q2 in the iterative offline counterpart but can not

Table 3: Comparison on D4RL *-v2.

| | | PLAS | DROP | DROP |
|---|---|---|---|---|
| antmaze | umaze | 70.7 | 78 | **80** |
| | umaze-diverse | 45.3 | 62 | **66** |
| | medium-play | 16 | **34** | 30 |
| | meidum-diverse | 0.7 | 24 | **30** |
| | large-play | 0.7 | 36 | **42** |
| | large-diverse | 0.3 | 20 | **26** |
| | | LAPO | DROP | DROP+ |
| medium | walker2d | 80.8 | 79.1 | **82.1** |
| | hopper | 51.6 | 59.5 | **61.5** |
| | halfcheetah | 46 | 43.1 | **52.4** |
| random | walker2d | 1.3 | 3 | **5.2** |
| | hopper | **23.5** | 5.5 | 20.8 |
| | halfcheetah | 30.6 | 2.3 | **32.0** |

provide the potential benefit (deployment adaptation) by answering Q3 in the non-iterative paradigm.

We provide the comparison results in Table 3. We can observe that DROP (-Best-Ada) and DROP (-Grad-Ada) consistently achieves better performance than PLAS on AntMaze-*-v2 tasks. On the Gym *-medium domain, DROP (-Grad-Ada) also performs better than LAPO. However, there is a big performance gap between DROP and LAPO on the *-random domain. We speculate that it is mainly caused by the decomposition rule. In our DROP implementation, we heuristically use return to conduct task decomposition (motivated by RvS-R (Emmons et al., 2021)), while LAPO and PLAS conduct decomposition (learning latent policy) automatically. Similarly, to bridge the gap, we also adopt CVAE to model the offline data and afterwards take the learned latent embedding in CVAE as the embedding of behaviors, instead of conducting return-guided task decomposition. We provide implementation details (DROP+CVAE) in Appendix C.4 and new results in Tables 2 and 3 (DROP+), where we can see such CVAE-based DROP implementation can bring a substantial performance improvement. Further, in Table 6 (Appendix C.4), we compare DROP+CVAE to IQL (Kostrikov et al., 2021b), consistently demonstrating the competitive empirical performance of DROP approach against state-of-art offline iterative/non-iterative baselines.

## 6 CONCLUSION

In this work, we introduce non-iterative bi-level offline RL and based on this paradigm, we raise three questions (Q1, Q2, and Q3). To answer that, we reframe the offline RL problem as one of MBO and learn a score model (A1), introduce embedding learning and conservative regularization (A2), and propose deployment adaptation in testing (A3). We evaluate DROP on various tasks, showing that DROP gains comparable or better performance compared to prior methods.

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

## A APPENDIX

Here we provide the content of the appendix:

## B DISCUSSION AND FUTURE WORK

**Limitations.** DROP also has several limitations. First, the offline data decomposition dominates the following bi-level optimization, and thus choosing a suitable decomposition rule is a crucial requirement for policy inference (see experimental analysis in Appendix C.2). An exciting direction for future work is to study generalized task decomposition rules (Rao et al., 2021). In Appendix C.4, we also exhibit a potential of such generalized task decomposition by introducing CVAE into DROP's implementation, and find such a combination (DROP + CVAE) can bring practical performance improvement. Second, we find that when the number of sub-tasks is too large, the inference is unstable, where adjacent checkpoint models exhibit larger variance in performance (such instability also exists in prior offline RL methods, discovered by Fujimoto & Gu (2021)). One natural approach to this instability is conducting online fine-tuning (see Appendix C.3 for our empirical studies).

Going forward, we believe our work suggests a feasible alternative for generalizable offline robotic learning: by decomposing a single robotic dataset into multiple subsets, offline policy inference can benefit from performing model-based optimization (MBO) and the joint deployment adaptation procedure.

**Social impact:** Beyond a general offline RL improvement, the authors do not foresee negative social impacts.

## C MORE EXPERIMENTS

### C.1 SINGLE BEHAVIOR POLICY VS. MULTIPLE BEHAVIOR POLICIES

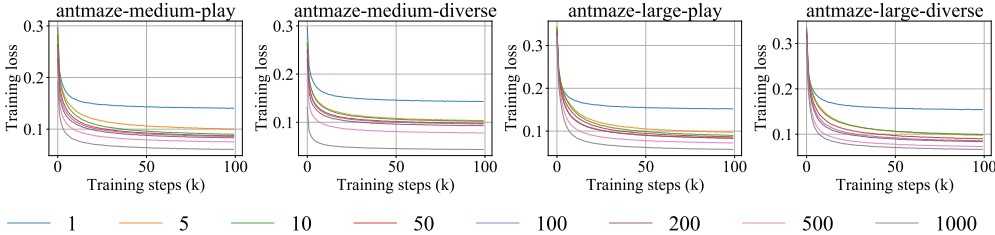

Figure 4: Learning curves of behavior cloning on AntMaze suites (*-v2) in D4RL, where the x-axis denotes the training steps, and the y-axis denotes the training loss. The number $N$ in the legend denotes the number of sub-tasks. If $N = 1$, we learn a single behavior policy for the whole offline dataset.

In Figure 4, we provide empirical evidence that learning a single behavior policy (using BC) is not sufficient to characterize the whole offline dataset, and multiple behavior policies (conducting task decomposition) deliver better resilience to characterize the offline data than a single behavior policy.

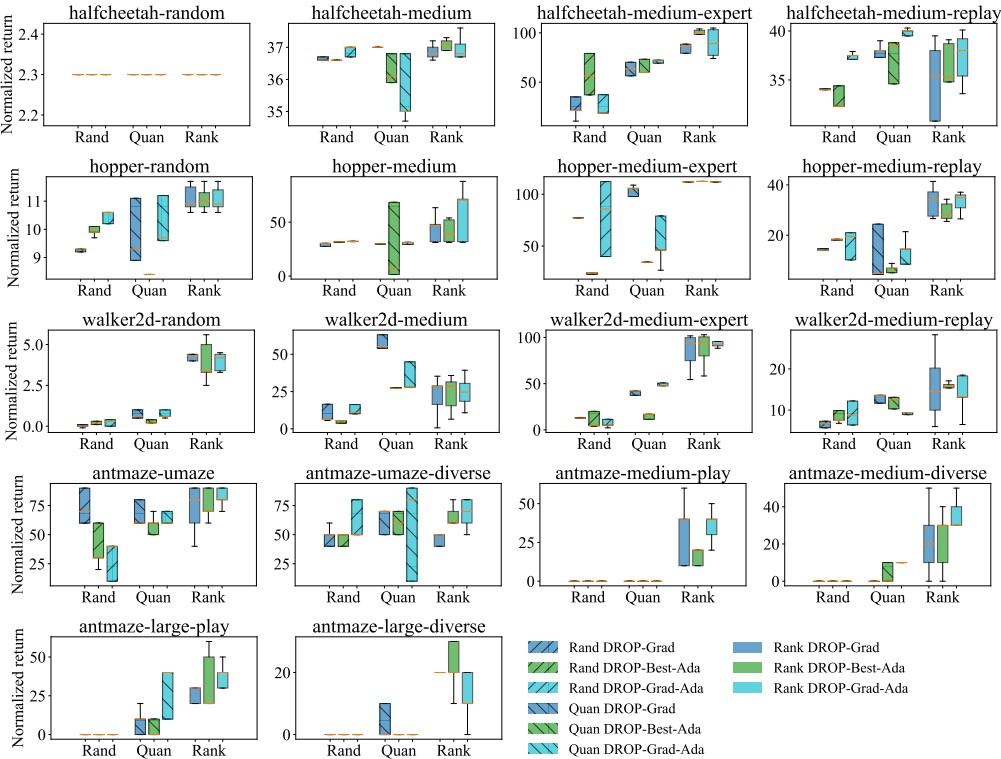

Figure 5: Comparison of three different decomposition rules on D4RL MuJoCo-Gym suite (*-v0) and AntMaze suite (*-v2), where "Rand", "Quan" and "Rank" denote the Random, Quantization, and Rank decomposition rules respectively. We can find across 18 tasks (AntMaze and MuJoCo-Gym suites) and 3 embedding inference methods (DROP-Grad, DROP-Best-Ada, and DROP-Grad-Ada), Rank is more stable and yields better performance compared with the other two decomposition rules.

## C.2 DECOMPOSITION RULES

In DROP algorithm, we explicitly decompose an offline task into multiple sub-tasks, over which we then reframe the offline policy learning problem as one of offline model-based optimization. In this section, we discuss three different designs for the task decomposition rule.

**Random**$(N, M)$**:** We decomposition offline dataset $\mathcal{D} := \{\tau\}$ into $N$ subsets, each of which contains at most $M$ trajectories that are randomly sampled from the offline dataset.

**Quantization**$(N, M)$**:** Leveraging the returns of trajectories in offline data, we first quantize offline trajectories into $N$ bins, and then randomly sample at most $M$ trajectories (as a sub-task) from each bin. Specifically, in the $i$-th bin, the quantized trajectories $\{\tau_i\}$ satisfy $\mathrm{R}_{\min} + \Delta * i < \mathrm{Return}(\tau_i) \leq \mathrm{R}_{\min} + \Delta * (i+1)$, where $\Delta = \frac{(\mathrm{R}_{\max} - \mathrm{R}_{\min})}{N}$, $\mathrm{Return}(\tau_i)$ denotes the return of trajectory $\tau_i$, and $\mathrm{R}_{\max}$ and $\mathrm{R}_{\min}$ denote the maximum and minimum trajectory returns in the offline dataset respectively.

**Rank**$(N, M)$**:** We first rank the offline trajectories descendingly based on their returns, and then sequentially sample $M$ trajectories for each subset. (*We adopt this decomposition rule in main paper.*)

In Figure 5, we provide the comparison of the above three decomposition rules (see the selected number of sub-tasks and the number of trajectories in each sub-task in Table 9). We can find that across a verity of tasks, decomposition rule has a fundamental impact on the subsequent model-based optimization. Across different tasks and different embedding inference rules, Random and Quantization decomposition rules tend to exhibit large performance fluctuations, which reveals the importance of choosing a suitable task decomposition rule. In our paper, we adopt the Rank decomposition rule, as it demonstrates a more robust performance shown in Figure 5. In Appendix C.4, we adopt the conditional variational auto-encoder (CVAE) to conduct automatic task decomposi-

Table 4: Comparison between our DROP (using the Rank decomposition rule) and filtered behavior cloning (F-BC) on D4RL AntMaze and MuJoCo suites (*-v2). We take the baseline results of BC and F-BC from Emmons et al. (2021), where F-BC is trained over the top $10\%$ trajectories, ordered by the returns. Our DROP results are computed over 5 seeds and 10 episodes for each seed.

| Tasks | | BC | F-BC | DROP-Grad | DROP-Best-Ada | DROP-Grad-Ada |
|---|---|---|---|---|---|---|
| antmaze-umaze | | 54.6 | 60 | $72 \pm 17.2$ | $78 \pm 11.7$ | $80 \pm 12.6$ |
| antmaze-umaze-diverse | | 45.6 | 46.5 | $48 \pm 22.3$ | $62 \pm 16$ | $66 \pm 12$ |
| antmaze-medium-play | | 0 | 42.1 | $24 \pm 10.2$ | $34 \pm 12$ | $30 \pm 21$ |
| antmaze-medium-diverse | | 0 | 37.2 | $20 \pm 19$ | $24 \pm 12$ | $30 \pm 16.7$ |
| antmaze-large-play- | | 0 | 28 | $24 \pm 8$ | $36 \pm 17.4$ | $42 \pm 17.2$ |
| antmaze-large-diverse | | 0 | 34.3 | $14 \pm 8$ | $20 \pm 14.1$ | $26 \pm 13.6$ |
| antmaze total | | 100.2 | 248.1 | 202 | 254 | **274** |
| halfcheetah | random | 2.3 | 2 | $2.3 \pm 0$ | $2.3 \pm 0$ | $2.3 \pm 0$ |
| hopper | random | 4.8 | 4.1 | $5.1 \pm 0.8$ | $5.4 \pm 0.7$ | $5.5 \pm 0.6$ |
| walker2d | random | 1.7 | 1.7 | $2.8 \pm 1.7$ | $3 \pm 1.6$ | $3 \pm 1.8$ |
| halfcheetah | medium | 42.6 | 42.5 | $42.4 \pm 0.7$ | $42.9 \pm 0.4$ | $43.1 \pm 0.4$ |
| hopper | medium | 52.9 | 56.9 | $57.5 \pm 6.4$ | $60.3 \pm 6.1$ | $59.5 \pm 5.1$ |
| walker2d | medium | 75.3 | 75 | $76.5 \pm 2.4$ | $75.8 \pm 3$ | $79.1 \pm 1.4$ |
| halfcheetah | medium-replay | 36.6 | 40.6 | $39.5 \pm 1$ | $40.4 \pm 0.8$ | $40.3 \pm 1.2$ |
| hopper | medium-replay | 18.1 | 75.9 | $48 \pm 17.7$ | $83.4 \pm 6.5$ | $87.4 \pm 2.1$ |
| walker2d | medium-replay | 26 | 62.5 | $37.4 \pm 13.5$ | $60.9 \pm 7.4$ | $61.9 \pm 2.3$ |
| halfcheetah | medium-expert | 55.2 | 92.9 | $86.6 \pm 3.9$ | $88.5 \pm 1.2$ | $88.9 \pm 2$ |
| hopper | medium-expert | 52.5 | 110.9 | $103.5 \pm 6.3$ | $102.5 \pm 6.2$ | $105.9 \pm 4.9$ |
| walker2d | medium-expert | 107.5 | 109 | $107.5 \pm 2$ | $106.8 \pm 3.9$ | $106.9 \pm 3.6$ |
| mujoco-gym total | | 475.5 | 674 | 609.1 | 672.2 | **683.8** |

tion (treating each trajectory in offline dataset as an individual task) and we find such implementation (DROP+CVAE) can further improve DROP's performance. In future work, we also encourage better decomposition rules to decompose offline tasks so as to enable more effective model-based optimization for offline RL tasks.

**Comparison with filtered behavior cloning.** We also note that the Rank decomposition rule leverages more high-quality trajectories than the other two decomposition rules (Random and Quantization). Thus, a natural question to ask is, is the performance of Rank better than that of Random and Quantization due to the presence of more high-quality trajectories in the decomposed sub-tasks? That is, whether DROP (using the Rank decomposition rule) only conducts behavioral cloning over those high-quality trajectories, thus leading to better performance.

To answer the above question, we compare DROP (using the Rank decomposition rule) with the filtered behavior cloning (F-BC), where the latter (F-BC) performs behavior cloning after filtering for trajectories with highest returns. We provide the comparison results in Table 4. We can find that in AntMaze tasks, the overall performance of DROP is higher than that of F-BC. For the MuJoCo-Gym suite, DROP-based methods outperforms F-BC on these offline tasks that contain a plenty of sub-optimal trajectories, including the random, medium, and medium-replay domains. This result indicates that DROP can leverage the sort of embedding inference (extrapolation) to find a better policy beyond all the behavior policies in sub-tasks, which is more effective than simply performing imitation learning on a subset of the dataset.

## C.3 ONLINE FINE-TUNING

**Online fine-tuning (checkpoint-level).** In Figure 6, we show the learning curves of DROP-Best on four DR4L tasks. We can find that DROP exhibits a high-variance (in performance) across

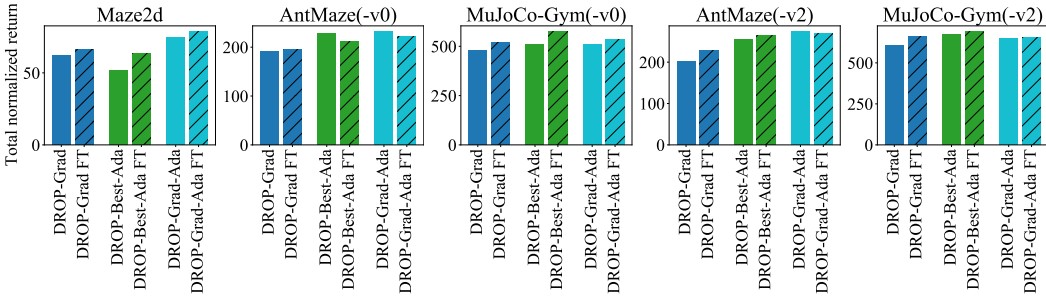

Figure 6: Learning curves of DROP, where the x-axis denotes the training steps (k), y-axis denotes the evaluation return (using DROP-Best embedding inference rule). We only show two seeds for legibility.

training steps[6], which means the performance of the agent may be dependent on the specific stopping point chosen for evaluation (such instability also exists in prior offline RL methods (Fujimoto & Gu, 2021)).

To choose a suitable stopping checkpoint over which we perform the DROP inference (DROP-Grad, DROP-Best-Ada and DROP-Grad-Ada), we propose to conduct *checkpoint-level* online fine-tuning (see Algorithm 3 in Section D for more details): we evaluate each of the latest $T$ checkpoint models and choose the best one that leads to the highest episode return.

In Figure 7, we show the total normalized returns across all the tasks in each suite (including Maze2d, AntMaze, and MuJoCo-Gym). We can find that in most tasks, fine-tuning (FT) can guarantee a performance improvement. However, we also find such fine-tuning causes negative impacts in performance in AntMaze(*-v0) suite. The main reason is that, in this checkpoint-level fine-tuning, we choose the "suitable" checkpoint model using the DROP-Best embedding inference rule, while we adopt the other three embedding inference rules (DROP-Grad, DROP-Best-Ada and DROP-Grad-Ada) at the test time. Such finding also implies that the success of DROP's deployment adaptation is not entirely dependent on the best embedding across sub-tasks [7] (*i.e.*, the best embedding $\mathbf{z}_0^*(\mathbf{s}_0)$ in DROP-Best), but requires switching between some "suboptimal" embeddings (using DROP-Best-Ada) or extrapolating new embeddings (using DROP-Grad-Ada).

Figure 7: Total normalized returns across all the tasks in Maze2d, AntMaze, and MuJoCo-Gym suites.

**Online fine-tuning (embedding-level).** Beyond the above checkpoint-level fine-tuning procedure, we can also conduct *embedding-level* online fine-tuning: we aim to choose a suitable gradient update step for the gradient-based embedding inference rules (including DROP-Grad and DROP-Grad-Ada). Similar to the checkpoint-level fine-tuning, we first conduct the deployment adaptation pro-

---

[6]In view of such instability, we evaluate our methods over multiple checkpoints for each seed, instead of choosing the final checkpoint models during the training loop (see the detailed evaluation protocol in Appendix D).

[7]Conversely, if the performance of DROP depends on the best embedding across sub-tasks (*i.e.*, $\mathbf{z}_0^*(\mathbf{s}_0)$ in DROP-Best), then the checkpoint model we choose by fine-tuning with DROP-Best should enable a consistent performance improvement for rules that perform embedding inference with DROP-Best-Ada and DROP-Grad-Ada. However, we find a performance drop in AntMaze(*-v0) suite, which means these is no explicit dependency between the best embedding $\mathbf{z}_0^*(\mathbf{s}_0)$ and the inferred embedding using the adaptive inference rules (DROP-*-Ada).

cedure (DROP-Grad and DROP-Grad-Ada) over a set of gradient update steps, and then choose the best step that leads to the highest episode return (see Algorithm 4 in Section D for more details).

In Table 5, we compare our DROP (DROP-Grad and DROP-Grad-Ada) to three offline RL methods (AWAC (Nair et al., 2020b), CQL (Kumar et al., 2020) and IQL (Kostrikov et al., 2021b)), reporting the initial performance and the performance after online fine-tuning. We can find that the embedding-level fine-tuning (0.3M) enables a significant improvement in performance. The fine-tuned DROP-Grad-Ada (0.3M) outperforms the AWAC and CQL counterparts in most tasks, even though we take less rollout steps to conduct the online fine-tuning (baselines take 1M online rollout steps, while DROP-based fine-tuning takes 0.3M steps). However, there is still a big gap between the fine-tuned IQL and the embedding-level fine-tuned DROP (0.3M). Considering that there remains 0.7M online steps in the comparison, we further conduct "parametric-level" fine-tuning (updating the parameters of the policy network) for our DROP-Grad-Ada on medium-* and large-* tasks, we can find which achieves competitive fine-tuning performance even compared with IQL.

Table 5: Online fine-tuning results (initial performance $\rightarrow$ performance after online fine-tuning). The baseline results of AWAC, CQL, and IQL are taken from Kostrikov et al. (2021b), where they run 1M online steps to fine-tune the learned policy. For our DROP method (DROP-Grad and DROP-Grad-Ada), we run 0.3M ($= 6_{\text{checkpoint}} \times 50_{K_{\max}} \times 1000_{\text{steps per episode}}$) online steps to fine-tune (embedding-level) the policy, *i.e.*, aiming to find the optimal gradient ascent step that is used to infer the contextual embedding $\mathbf{z}^*(\mathbf{s}_0)$ or $\mathbf{z}^*(\mathbf{s}_t)$ for $\pi^*(\mathbf{a}_t|\mathbf{s}_t) := \beta(\mathbf{a}_t|\mathbf{s}_t, \cdot)$ (see Algorithm 4 for the details). Moreover, for medium-* and large-* tasks, we conduct additional parametric-level fine-tuning, with 0.7M online steps to update the policy's parameters. Our DROP results are computed over 5 seeds and 10 episodes for each seed.

| Task (*-v0) | AWAC | CQL | IQL | DROP-Grad | DROP-Grad-Ada | |
|---|---|---|---|---|---|---|
| umaze | $56.7 \rightarrow 59$ | $70.1 \rightarrow 99.4$ | $86.7 \rightarrow 96$ | $70 \rightarrow 96 \pm 1.2$ | $76 \rightarrow 98 \pm 0$ | |
| umaze-diverse | $49.3 \rightarrow 49$ | $31.1 \rightarrow 99.4$ | $75 \rightarrow 84$ | $54 \rightarrow 88 \pm 8$ | $66 \rightarrow 94 \pm 4.9$ | |
| medium-play | $0 \rightarrow 0$ | $23 \rightarrow 0$ | $72 \rightarrow 95$ | $20 \rightarrow 56 \pm 8.9$ | $30 \rightarrow 50 \pm 6.3$ | $\rightarrow 94 \pm 2.9$ |
| medium-diverse | $0.7 \rightarrow 0.3$ | $23 \rightarrow 32.3$ | $68.3 \rightarrow 92$ | $12 \rightarrow 44 \pm 4.9$ | $22 \rightarrow 38 \pm 4.9$ | $\rightarrow 96 \pm 0.8$ |
| large-play | $0 \rightarrow 0$ | $1 \rightarrow 0$ | $25.5 \rightarrow 46$ | $16 \rightarrow 38 \pm 8.9$ | $16 \rightarrow 40 \pm 6.3$ | $\rightarrow 53 \pm 1.3$ |
| large-diverse | $1 \rightarrow 0$ | $1 \rightarrow 0$ | $42.6 \rightarrow 60.7$ | $20 \rightarrow 40 \pm 13.6$ | $22 \rightarrow 46 \pm 10.2$ | $\rightarrow 58 \pm 4.5$ |
| | $\underbrace{\rightarrow}_{1M}$ | $\underbrace{\rightarrow}_{1M}$ | $\underbrace{\rightarrow}_{1M}$ | $\underbrace{\rightarrow}_{0.3M}$ | $\underbrace{\rightarrow}_{0.3M}$ | $\underbrace{\rightarrow}_{0.7M}$ |

## C.4 DROP + CVAE

**CVAE-based embedding learning.** Similar to LAPO (Chen et al., 2022) and PLAS (Zhou et al., 2020), we adopt the conditional variational auto-encoder (CVAE) to model offline data. Specifically, we learn the contextual policy and behavior embedding:

$$\beta(\mathbf{a}|\mathbf{s}, \mathbf{z}), \phi(\mathbf{z}|\mathbf{s}) \leftarrow \arg\max_{\beta, \phi} \mathbb{E}_{(\mathbf{s},\mathbf{a})\sim\mathcal{D}}\mathbb{E}_{(\mathbf{z})\sim\phi(\mathbf{z}|\mathbf{s})}\big[\log\beta(\mathbf{a}|\mathbf{s}, \mathbf{z})\big] - \mathrm{KL}(\phi(\mathbf{z}|\mathbf{s})\|p(\mathbf{z})). \quad (10)$$

Then, we learn the score model $f$ with the TD-error and the conservative regularization:

$$f \leftarrow \arg\min_f \mathbb{E}_{(\mathbf{s},\mathbf{a},\mathbf{s}',\mathbf{a}')\sim\mathcal{D}}\left[\big(R(\mathbf{s}, \mathbf{a}) + \gamma\bar{f}(\mathbf{s}', \mathbf{a}', \phi(\mathbf{z}|\mathbf{s})) - f(\mathbf{s}, \mathbf{a}, \phi(\mathbf{z}|\mathbf{s}))\big)^2\right], \quad (11)$$

$$\text{s.t. } \mathbb{E}_{\mathbf{s}\sim\mathcal{D},\mathbf{z}\sim\mu(\mathbf{z}),\mathbf{a}\sim\beta(\mathbf{a}|\mathbf{s},\mathbf{z})}[f(\mathbf{s}, \mathbf{a}, \mathbf{z})] - \mathbb{E}_{\mathbf{s}\sim\mathcal{D},\mathbf{z}\sim\phi(\mathbf{z}|\mathbf{s}),\mathbf{a}\sim\beta(\mathbf{a}|\mathbf{s},\mathbf{z})}[f(\mathbf{s}, \mathbf{a}, \mathbf{z})] \leq \eta,$$

where $\bar{f}$ denotes a target network and $\mu(\mathbf{z})$ denotes the uniform distribution over the $\mathcal{Z}$-space.

In testing, we also dynamically adapt the outer-level optimization, setting policy inference with $\pi^*(\mathbf{a}|\mathbf{s}) = \beta(\mathbf{a}|\mathbf{s}, \mathbf{z}^*(\mathbf{s}))$, where $z^*(\mathbf{s}) = \arg\max_z f(\mathbf{s}, \beta(\mathbf{a}|\mathbf{s}, \mathbf{z}), \mathbf{z})$.

In Table 6, we compare DROP+CVAE (-Grad-Ada) with LAPO (Chen et al., 2022), PLAS (Zhou et al., 2020), CQL Kumar et al. (2020), IQL Kostrikov et al. (2021b) and the naive implementation of DROP(-Grad-Ada) (conducting return-guided task decomposition and afterward learning behavior embedding as in Equation 6). We highlight that even there is a big performance gap between DROP and baselines (LAPO and PLAS) in Gym-MuJoCo *-random tasks, our CVAE-based implementation (DROP+CVAE) can bridge such performance gap. Further, in *-medium tasks, DROP+CVAE

Table 6: Comparison (on D4RL benchmark) between DROP (including the implementation of return-guided task decomposition and the implementation CVAE-based embedding learning), latent policy baselines (LAPO and PLAS) and other two representative baselines (CQL and IQL).

|  |  | LAPO | PLAS | CQL | IQL | DROP | DROP+CVAE |
|---|---|---|---|---|---|---|---|
| antmaze | umaze | — | 70.7 | 74.0 | 87.5 | 80 | 90.5 ± 2.4 |
|  | umaze-diverse | 91.3 | 45.3 | 84.0 | 62.2 | 66 | 85.1 ± 7.8 |
|  | medium-play | — | 16 | 61.2 | 71.2 | 30 | 68.2 ± 16.5 |
|  | medium-diverse | 85.7 | 0.7 | 53.7 | 70.0 | 30 | 75.4 ± 9.4 |
|  | large-play | — | 0.7 | 15.8 | 39.6 | 42 | 50.1 ± 13.6 |
|  | large-diverse | 61.7 | 0.3 | 14.9 | 47.5 | 26 | 52.2 ± 12.0 |
| random | walker2d-random | 1.3 | 3.1 | -0.23 | 5.4 | 3 | 5.2 ± 1.6 |
|  | hopper-random | 23.5 | 10.5 | 8.3 | 7.9 | 5.5 | 20.8 ± 0.3 |
|  | halfcheetah-random | 30.6 | 25.8 | 22.2 | 13.1 | 2.3 | 32.0 ± 2.5 |
| medium | walker2d-medium | 80.8 | 44.6 | 82.1 | 77.9 | 79.1 | 82.1 ± 5.2 |
|  | hopper-medium | 51.6 | 32.9 | 71.6 | 65.8 | 59.5 | 61.5 ± 3.7 |
|  | halfcheetah-medium | 46 | 39.3 | 49.8 | 47.8 | 43.1 | 52.4 ± 2.2 |

can further improve the performance (of DROP), surpassing both LAPO and PLAS that similarly adopt the CVAE to model the offline data. Besides, we compare DROP+CVAE to baselines CQL and IQL, where we see DROP+CVAE can also achieve a competitive performance.

### C.5 ABLATION STUDY

Note that our embedding inference depends on the learned score model $f$. Without proper regularization, such inference will lead to out-of-distribution embeddings that are erroneously high scored (Q2). Here we conduct an ablation study to examine the impact of the conservative regularization used for learning the score model.

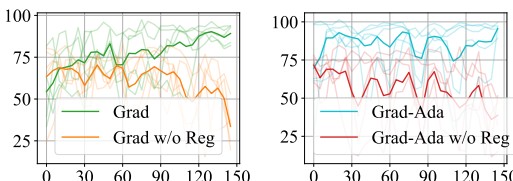

In Figure 8, we compare DROP-Grad and DROP-Grad-Ada to their naive implementation (*w.o. Reg*) that ablates the regularization on halfcheetah-medium-expert. We can find that removing the conservative regularization leads to unstable performance when changing the update steps of gradient-based optimization. However, we empirically find that in

Figure 8: Ablation on the conservative regularization. The y-axis represents the normalize return, and the x-axis represents the number of gradient-ascent steps used for embedding inference at deployment. We plot each random seed as a transparent line; the solid line corresponds to the average across 5 seeds.

some tasks such naive implementation (*w/o Reg*) does not necessarily bring unstable inference (Appendix E). Although improper gradient update step leads to faraway embeddings, to some extent, embedding-conditioned behavior policy can correct such deviation.

## D IMPLEMENTATION DETAILS

For the practical implementation of DROP, we parameterize the task embedding function $\phi(\mathbf{z}|n)$, the contextual behavior policy $\beta(\mathbf{a}|\mathbf{s},\mathbf{z})$ and the score model $f(\mathbf{s},\mathbf{a},\mathbf{z})$ with neural networks (see Appendix D for specific architectures). For Equation 8, we construct a Lagrangian and solve the optimization through primal-dual gradient descent. For the choice of $\mu(\mathbf{z})$, we simply set $\mu(\mathbf{z})$ to be the uniform distribution over the $\mathcal{Z}$-space and empirically find that such uniform sampling can effectively avoid the out-of-distribution extrapolation at inference.

**Lagrangian Relaxation.** To optimize the constrained objective in Equation 8 in the main paper, we construct a Lagrangian and solve the optimization through primal-dual gradient descent,

$$\min_{f} \max_{\lambda > 0} \quad \mathbb{E}_{\mathcal{D}_n \sim \mathcal{D}_{[N]}} \mathbb{E}_{(\mathbf{s},\mathbf{a},\mathbf{s}',\mathbf{a}') \sim \mathcal{D}_n} \left[ \left( R(\mathbf{s},\mathbf{a}) + \gamma \bar{f}(\mathbf{s}',\mathbf{a}',\phi(\mathbf{z}|n)) - f(\mathbf{s},\mathbf{a},\phi(\mathbf{z}|n)) \right)^2 \right] +$$

$$\lambda \left( \mathbb{E}_{n,\mu(\mathbf{z})} \mathbb{E}_{\mathbf{s} \sim \mathcal{D}_n, \mathbf{a} \sim \beta(\mathbf{a}|\mathbf{s},\mathbf{z})} \left[ f(\mathbf{s},\mathbf{a},\mathbf{z}) \right] - \mathbb{E}_{n,\phi(\mathbf{z}|n)} \mathbb{E}_{\mathbf{s} \sim \mathcal{D}_n, \mathbf{a} \sim \beta(\mathbf{a}|\mathbf{s},\mathbf{z})} \left[ f(\mathbf{s},\mathbf{a},\mathbf{z}) \right] - \eta \right).$$

This unconstrained objective implies that if the expected difference in scores of out-of-distribution embeddings and in-distribution embeddings is less than a threshold $\eta$, $\lambda$ is going to be adjusted to 0, on the contrary, $\lambda$ is likely to take a larger value, used to punish the over-estimated value function. This objective encourages that out-of-distribution embeddings score lower than in-distribution embeddings, thus performing embedding inference will not lead to these out-of-distribution embeddings that are falsely and over-optimistically scored by the learned score model.

In our experiments, we tried five different values for the Lagrange threshold $\eta$ (1.0, 2.0, 3.0, 4.0 and 5.0). We did not observe a significant difference in performance across these values. Therefore, we simply set $\eta = 2.0$.

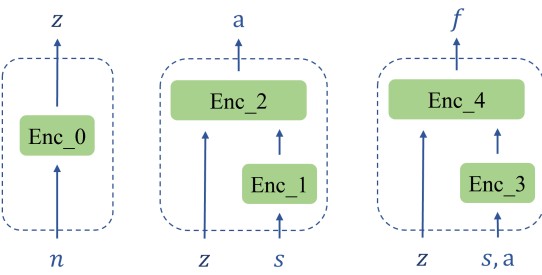

Figure 9: Architectures of the task embedding network $\phi(\mathbf{z}|\mathbf{s})$, the contextual behavior policy $\beta(\mathbf{a}|\mathbf{s},\mathbf{z})$, and the score function $f(\mathbf{s},\mathbf{a},\mathbf{z})$ (from left to right).

**Hyper-parameters.** In Figure 9, we provide the network architecture of the task embedding $\phi(\mathbf{z}|\mathbf{s})$, the contextual behavior policy $\beta(\mathbf{a}|\mathbf{s},\mathbf{z})$, and the score function $f(\mathbf{s},\mathbf{a},\mathbf{z})$, where the corresponding hyper-parameters are provided in Table 7. For the gradient ascent update steps (used for embedding inference), we set $K = 100$ for all the embedding inference rules in experiments.

Table 7: Hyper-parameters.

|  | Enc_0 | Enc_1 | Enc_2 | Enc_3 | Enc_4 |
|---|---|---|---|---|---|
| Optimizer | Adam | Adam | Adam | Adam | Adam |
| Hidden layer | 2 | 2 | 3 | 2 | 3 |
| Hidden dim | 512 | 512 | 512 | 512 | 512 |
| Activation function | ReLU | ReLU | ReLU | ReLU | ReLU |
| Learning rate | 1.00E-03 | 1.00E-03 | 1.00E-03 | 1.00E-03 | 1.00E-03 |
| Mini-batch size | 1024 | 1024 | 1024 | 1024 | 1024 |

In Table 9, we provide the number of sub-tasks, the number of trajectories in each sub-task, and the dimension of the embedding for each sub-task (behavior policy). The selection of hyperparameter N is based on two evaluation metrics: (1) the fitting loss of the decomposed behavioral policies to the offline data, and (2) the testing performance of DROP. Specifically,

- (Step1) Over a hyperparameter (the number of sub-tasks) set, we conduct the hyperparameter search using the fitting loss of behavior policies, then we choose/filter the four best hyperparameters;
- (Step2) We follow the normal practice of hyperparameter selection and tune the four hypermeters selected in Step1 by interacting with the simulator to estimate the performance of DROP under each hyperparameter setting.

---

**Algorithm 3** DROP: Online fine-tuning (checkpoint-level)

---

**Require:** Env, last $T$ checkpoint models: $\beta_t(\mathbf{a}|\mathbf{s}, \mathbf{z})$ and $f_t(\mathbf{s}, \mathbf{a}, \mathbf{z})$ $(t = 1, \cdots, T)$.

1: $R_{\text{MAX}} = -\infty$.
2: $\beta_{\text{best}} \leftarrow$ None.
3: $f_{\text{best}} \leftarrow$ None.
4: **while** $t = 1, \cdots, T$ **do**
5:    $\mathbf{s}_0 =$ Env.Reset().
6:    $\mathbf{z}_0^*(\mathbf{s}_0) \leftarrow$ Conduct embedding inference with DROP-Best.
7:    Return $\leftarrow$ Evaluate $\beta_t$ and $f_t$ on Env, setting $\pi^*(\mathbf{a}|\mathbf{s}) = \beta(\mathbf{a}|\mathbf{s}, \mathbf{z}_0^*(\mathbf{s}_0))$.
8:    **if** $R_{\text{MAX}} <$ Return **then**
9:       Update the best checkpoint models: $\beta_{\text{best}} \leftarrow \beta_t$, $f_{\text{best}} \leftarrow f_t$.
10:      Update the optimal return: $R_{\text{MAX}} \leftarrow$ Return.
11:   **end if**
12: **end while**

**Return:** $\beta_{\text{best}}$ and $f_{\text{best}}$.

---

We provide the hyperparameter sets in Table 8. In Step2, we tune the (filtered) hyperparameters using 1 seed, then evaluate the best hyperparameter by training on an additional 4 seeds and finally report the results on the 5 total seeds (see next "evaluation protocol"). In Antmaze domain, a single

Table 8: Hyperparameter (the number of sub-tasks) set.

| tasks | the number of sub-tasks |
|---|---|
| Antmaze | 500 (v0), 150 (v2) |
| Gym-mujoco | 10, 20, 50, 100, 200, 500, 800, 1000 |
| Adroit | 10, 20, 50, 100, 200, 500, 800, 1000 |

fixed N works well for many tasks; while in Gym-mujoco and Adroit domains, we did not find a fixed N that provides good results for all tasks in the corresponding domain in D4RL, thus we use the above hyperparameter selection rules (Step1 and Step2) to choose the number N.

**Environment details.** For the comparison of our method to prior iterative offline RL methods, we consider the v0 versions of the datasets in D4RL[8]. We take the baseline results of BEAR, BCQ, CQL, and BRAC-p from the D4RL paper (Fu et al., 2020), and take the results of TD3+BC from their origin paper (Fujimoto & Gu, 2021). For the comparison of our method to prior non-iterative offline RL method, we use the v2 versions of the dataset in D4RL. All the baseline results of behavior cloning (BC), Decision Transform (DT), RvS-R, and Onestep are taken from Emmons et al. (2021). In our implementation of COMs, we take the parameters (neural network weights) of behavior policies as the design input for the score model; and during testing, we conduct parameters inference (outer-level optimization) with 200 steps gradient ascent over the learned score function, then the rollout policy is initialized with the inferred parameters. For the specific architecture, we instance the policy network with $\dim(\mathcal{S})$ input units, two layers with 64 hidden units, and a final output layer with $\dim(\mathcal{A})$.

**Evaluation protocol.** We evaluate our results over 5 seeds. For each seed, instead of taking the final checkpoint model produced by a training loop, we take the last $T$ ($T = 6$ in our experiments) checkpoint models, and evaluate them over 10 episodes for each checkpoint. That is to say, we report the average of the evaluation scores over $5_{\text{seed}} \times 6_{\text{checkpoint}} \times 10_{\text{episode}}$ rollouts.

*Online fine-tuning (checkpoint-level):* Instead of re-training the learned (final) policy with online rollouts, we fine-tune our policy with enumerated trail-and-error over the last $T$ checkpoint models (Algorithm 3). Specifically, for each seed, we run the last $T$ checkpoint models in environment over one episode for each checkpoint. The checkpoint model which achieves the maximum episode return is returned. In essence, this fine-tuning procedure imitates the online RL evaluation protocol: if the current policy is unsatisfactory, we can use checkpoints of previous iterations of the policy.

*Online fine-tuning (embedding-level):* The embedding-level fine-tuning aims to find a suitable gradient ascent step that is used to conduct the embedding inference in DROP-Grad *or* DROP-Grad-Ada.

---

[8] We noticed that Maze2D-v0 in the D4RL dataset (https://rail.eecs.berkeley.edu/datasets/) is not available, so we used v1 version instead in our experiment. For simplicity, we still use v0 in the paper exposition.

---

**Algorithm 4** DROP: Online fine-tuning (embedding-level)

---

**Require:** Env, last $T$ checkpoint models: $\beta_t(\mathbf{a}|\mathbf{s}, \mathbf{z})$ and $f_t(\mathbf{s}, \mathbf{a}, \mathbf{z})$ ($t = 1, \cdots, T$).

1: $R_{\text{MAX}} = -\infty$.
2: $\beta_{\text{best}} \leftarrow$ None.
3: $f_{\text{best}} \leftarrow$ None.
4: $k_{\text{best}} \leftarrow 0$.
5: **while** $t = 1, \cdots, T$ **do**
6:     **while** $k = 1, \cdots, K_{\text{max}}$ **do**
7:         $\mathbf{s}_0 =$ Env.Reset().
        # Conduct embedding inference with DROP-Grad *or* DROP-Grad-Ada
8:         Return $\leftarrow$ Evaluate $\beta_t$ and $f_t$ on Env, setting $\pi^*(\mathbf{a}|\mathbf{s}) = \beta(\mathbf{a}|\mathbf{s}, \mathbf{z}^*(\mathbf{s}_0))$ *or* $\beta(\mathbf{a}|\mathbf{s}, \mathbf{z}^*(\mathbf{s}))$,
        where we conduct $k$ gradient ascent steps to obtain $\mathbf{z}^*(\mathbf{s}_0)$ *or* $\mathbf{z}^*(\mathbf{s})$.
9:         **if** $R_{\text{MAX}} <$ Return **then**
10:             Update the best checkpoint models: $\beta_{\text{best}} \leftarrow \beta_t$, $f_{\text{best}} \leftarrow f_t$.
11:             Update the best gradient update step: $k_{\text{best}} \leftarrow k$.
12:             Update the optimal return: $R_{\text{MAX}} \leftarrow$ Return.
13:         **end if**
14:     **end while**
15: **end while**

**Return:** $\beta_{\text{best}}$, $f_{\text{best}}$ and $k_{\text{best}}$.

---

Thus, we enumerate a list of gradient update steps and pick the best update step (according to the episode returns).

**Codebase.** Our code is based on d3rlpy: `https://github.com/takuseno/d3rlpy`. We provide our source code in the supplementary material.

**Computational resources.** The experiments were run on a computational cluster with 22x GeForce RTX 2080 Ti, and 4x NVIDIA Tesla V100 32GB for 20 days.

Table 9: The number ($N$) of sub-tasks, the number ($M$) of trajectories in each sub-task, and the dimension ($\dim(\mathbf{z})$) of the embedding for each sub-task.

| Domain | Task Name | Parameters (*-v0) | | | Parameters (*-v2) | | |
|---|---|---|---|---|---|---|---|
| | | $N$ | $M$ | $\dim(\mathbf{z})$ | $N$ | $M$ | $\dim(\mathbf{z})$ |
| Maze 2D | umaze | 500 | 5 | 5 | | | |
| | medium | 150 | 50 | 5 | | | |
| | large | 100 | 15 | 5 | | | |
| Antmaze | umaze | 500 | 50 | 5 | 150 | 50 | 5 |
| | umaze-diverse | 500 | 50 | 5 | 150 | 50 | 5 |
| | Medium-play | 500 | 50 | 5 | 150 | 50 | 5 |
| | Medium-diverse | 500 | 50 | 5 | 150 | 50 | 5 |
| | Large-play | 500 | 50 | 5 | 150 | 50 | 5 |
| | Large-diverse | 500 | 50 | 5 | 150 | 50 | 5 |
| halfcheetah | random | 1000 | 1 | 5 | 1000 | 1 | 5 |
| | medium | 100 | 2 | 5 | 100 | 2 | 5 |
| | medium-expert | 1000 | 1 | 5 | 1000 | 1 | 5 |
| | medium-replay | 50 | 10 | 5 | 50 | 10 | 5 |
| hopper | random | 100 | 2 | 5 | 100 | 2 | 5 |
| | medium | 100 | 5 | 5 | 100 | 5 | 5 |
| | medium-expert | 100 | 2 | 5 | 100 | 2 | 5 |
| | medium-replay | 50 | 5 | 5 | 10 | 30 | 5 |
| walker2d | random | 500 | 2 | 5 | 500 | 2 | 5 |
| | medium | 50 | 5 | 5 | 50 | 5 | 5 |
| | medium-expert | 50 | 5 | 5 | 50 | 5 | 5 |
| | medium-replay | 1000 | 5 | 5 | 10 | 50 | 5 |
| door | cloned | 1000 | 2 | 5 | | | |
| | expert | 500 | 5 | 5 | | | |
| | human | 50 | 3 | 5 | | | |
| hammer | cloned | 1000 | 1 | 5 | | | |
| | expert | 500 | 5 | 5 | | | |
| | human | 20 | 3 | 5 | | | |
| pen | cloned | 500 | 5 | 5 | | | |
| | expert | 500 | 5 | 5 | | | |
| | human | 50 | 5 | 5 | | | |
| relocate | cloned | 500 | 5 | 5 | | | |
| | expert | 500 | 5 | 5 | | | |
| | human | 50 | 4 | 5 | | | |

# E ADDITIONAL RESULTS

**Comparison with iterative offline RL baselines.** Here, we compare the performance of DROP (Grad, Best-Ada, and Grad-Ada ) to iterative offline RL baselines (BEAR (Kumar et al., 2019a), BCQ (Fujimoto et al., 2019), CQL (Kumar et al., 2020), BRAC-p (Wu et al., 2019), and TD3+BC (Fujimoto & Gu, 2021)) that perform iterative bi-level offline RL paradigm with (explicit or implicit) value/policy regularization in inner-level. In Table 10, we present the results for the Maze2D, AntMaze, Gym-MuJoCo, and Adroit suites in standard D4RL benchmark (*-v0), where we can find that DROP-Grad-Ada performs comparably or surpasses prior iterative bi-level works on most tasks: outperforming (or comparing) these policy regularized methods (BRAC-p and TD3+BC) on 25 out of 33 tasks and outperforming (or comparing) these value regularized algorithms (BEAR, BCQ, and CQL) on 19 out of 33 tasks.

**Comparison with RvS baselines.** As a complement to the results shown in Table 2 in the main paper, we provide the performance comparison for more tasks (Gym-MuJoCo and AntMaze *-v2 suites in D4RL) in Figure 11. We can find similar results as presented in our main paper: DROP consistently outperforms baselines in AntMaze tasks (the last three rows of sub-figures in Figure 11) and reaches comparable results on most tasks in Gym-MuJoCo suite.

Across different environments, we also find DROP exhibits more robust performance. Although baseline Onestep shows impressive performance in Gym-MuJoCo tasks, we can see that Onestep fails to make progress in AntMaze-medium-* and AntMaze-large-* tasks. However, we find that DROP-based methods exhibit a significant performance improvement in this AntMaze suite. We attribute the success of DROP outperforming Onestep (conducting only behavior policy improvement) to three advantages: (1) DROP learns *multiple* behavior policies; (2) DROP conducts policy improvement (corresponding to the embedding inference procedure) over *multiple* behavior policies; (3) DROP permits *deployment adaptation*, enabling the agent to "switch" behavior policies.

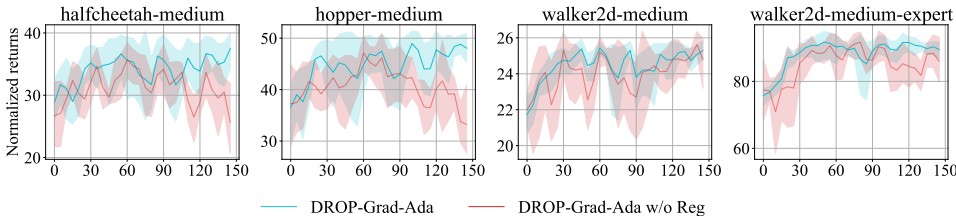

Figure 10: The performance comparison of DROP-Grad-Ada and DROP-Grad-Ada w/o Reg, where we ablate the conservative regularization for the *w/o Reg* implementation. The y-axis denotes the normalized return, the x-axis denotes the number of gradient-ascent steps used for embedding inference at deployment.

**Ablation studies.** In Figure 10, we provide more results for the ablation of the conservative regularization term in Equation 8 in the main paper. We can find that for the halfcheetah-medium and hopper-medium tasks, the performance of DROP-Grad-Ada w/o Reg depends on the choice of the gradient update steps, showing that too small or too large number of gradient update step deteriorates the performance. Such result is also consistent with COMs (Trabucco et al., 2021), which also observes the sensitivity of naive gradient update (*i.e.*, w/o Reg) to number of update steps used for design input inference. By comparison, conservative score model learned with DROP-Grad-Ada exhibits more stable and robust to the gradient update steps.

Further, we also find that in walker2d-medium and walker2d-medium-expert tasks, the naive gradient update (*w/o Reg*) does not affect performance significantly across a wide range of gradient update steps. The main reason is that although the excessive gradient updates lead to faraway embeddings, conditioned on the inferred embeddings, the learned contextual behavior policy can safeguard against the embeddings distribution shift. Compared to prior model-based optimization that conducts direct gradient optimization (inference) over the design input itself, such "self-safeguard" is a special merit in the offline RL domain as long as we reframe the offline RL problem as one of model-based optimization and conduct inference over the embedding space. Thus, we encourage the research community to pursue further study to this model-based optimization view for the offline RL problem.

**DROP results.** In Table 11, we provide our complete results (including the variance) on all tasks in the paper.

Table 10: Comparison of our method to prior offline methods that perform iterative (regularized) RL paradigm on D4RL. We take the baseline results of BEAR, BCQ, CQL and BRAC-p from Fu et al. (2020), and the results of TD3-BC from Fujimoto & Gu (2021). For all results of our method (DROP), we average the normalized returns across 5 seeds; for each seed, we run 10 evaluation episodes. For proper comparison, we use ▲ and ▲ to denote DROP (*-Ada) achieves *comparable or better* performance compared with value and policy regularized offline RL methods respectively.

| | Task Name | Value Reg. | | | Policy Reg. | | DROP- | | |
|---|---|---|---|---|---|---|---|---|---|
| | | BEAR | BCQ | CQL | BRAC-p | TD3+BC | Grad | Best-Ada | Grad-Ada |
| maze2d | umaze | 3.4 | 12.8 | 5.7 | 4.7 | – | 18.6 | 18.7 | 21.3▲▲ |
| | medium | 29.0 | 8.3 | 5.0 | 32.4 | – | 17.0 | 18.1 | 24.3 |
| | large | 4.6 | 6.2 | 12.5 | 10.4 | – | 26.7 | 14.7 | 28.8▲▲ |
| antmaze | umaze | 73.0 | 78.9 | 74.0 | 50.0 | – | 72.0 | 78.0▲▲ | 80.0▲▲ |
| | umaze-diverse | 61.0 | 55.0 | 84.0 | 40.0 | – | 48.0 | 62.0 ▲ | 66.0 ▲ |
| | medium-play | 0.0 | 0.0 | 61.2 | 0.0 | – | 24.0 | 34.0 ▲ | 30.0 ▲ |
| | medium-diverse | 8.0 | 0.0 | 53.7 | 0.0 | – | 20.0 | 24.0 ▲ | 30.0 ▲ |
| | large-play | 0.0 | 6.7 | 15.8 | 0.0 | – | 24.0 | 36.0▲▲ | 42.0▲▲ |
| | large-diverse | 0.0 | 2.2 | 14.9 | 0.0 | – | 14.0 | 20.0▲▲ | 26.0▲▲ |
| halfcheetah | random | 25.1 | 2.2 | 35.4 | 24.1 | 10.2 | 2.3 | 2.3 | 2.3 |
| | medium | 41.7 | 40.7 | 44.4 | 43.8 | 42.8 | 42.4 | 42.9▲▲ | 43.1▲▲ |
| | medium-expert | 53.4 | 64.7 | 62.4 | 44.2 | 97.9 | 86.6 | 88.5▲ | 88.9▲ |
| | medium-replay | 38.6 | 38.2 | 46.2 | 45.4 | 43.3 | 39.5 | 40.4 | 40.3 |
| hopper | random | 11.4 | 10.6 | 10.8 | 11.0 | 11.0 | 5.1 | 5.4 | 5.5 |
| | medium | 52.1 | 54.5 | 58.0 | 32.7 | 99.5 | 57.5 | 60.3▲ | 59.5▲ |
| | medium-expert | 96.3 | 110.9 | 98.7 | 1.9 | 112.2 | 103.5 | 102.5 | 105.9▲ |
| | medium-replay | 33.7 | 33.1 | 48.6 | 0.6 | 31.4 | 48.0 | 83.4▲▲ | 87.4▲▲ |
| walker2d | random | 7.3 | 4.9 | 7.0 | -0.2 | 1.4 | 2.8 | 3.0 ▲ | 3.0 ▲ |
| | medium | 59.1 | 53.1 | 79.2 | 77.5 | 79.7 | 76.5 | 75.8▲▲ | 79.1▲▲ |
| | medium-expert | 40.1 | 57.5 | 111.0 | 76.9 | 101.1 | 107.5 | 106.8▲▲ | 106.9▲▲ |
| | medium-replay | 19.2 | 15.0 | 26.7 | -0.3 | 25.2 | 37.4 | 60.9▲▲ | 61.9▲▲ |
| door | cloned | -0.1 | 0.0 | 0.4 | -0.1 | – | 0.5 | 2.5 | 2.7▲▲ |
| | expert | 103.4 | 99.0 | 101.5 | -0.3 | – | 98.6 | 102.2▲▲ | 102.6▲▲ |
| | human | -0.3 | 0.0 | 9.9 | -0.3 | – | 3.3 | 1.9 ▲ | 3.0 ▲ |
| hammer | cloned | 0.3 | 0.4 | 2.1 | 0.3 | – | 0.3 | 0.3 ▲ | 0.3 ▲ |
| | expert | 127.3 | 107.2 | 86.7 | 0.3 | – | 65.7 | 73.3 ▲ | 77.7 ▲ |
| | human | 0.3 | 0.5 | 4.4 | 0.3 | – | 1.1 | 0.3 | 2.1 ▲ |
| pen | cloned | 26.5 | 44.0 | 39.2 | 1.6 | – | 76.7 | 77.1▲▲ | 82.4▲▲ |
| | expert | 105.9 | 114.9 | 107.0 | -3.5 | – | 113.1 | 118.6▲▲ | 116.7▲▲ |
| | human | -1.0 | 68.9 | 37.5 | 8.1 | – | 71.1 | 85.2▲▲ | 81.5▲▲ |
| relocate | cloned | -0.3 | -0.3 | -0.1 | -0.3 | – | 0.1 | 0.5▲▲ | 0.2▲▲ |
| | expert | 98.6 | 41.6 | 95.0 | -0.3 | – | 2.5 | 6.2 ▲ | 5.4 ▲ |
| | human | -0.3 | -0.1 | 0.2 | -0.3 | – | 0.0 | 0.0 | 0.0 |

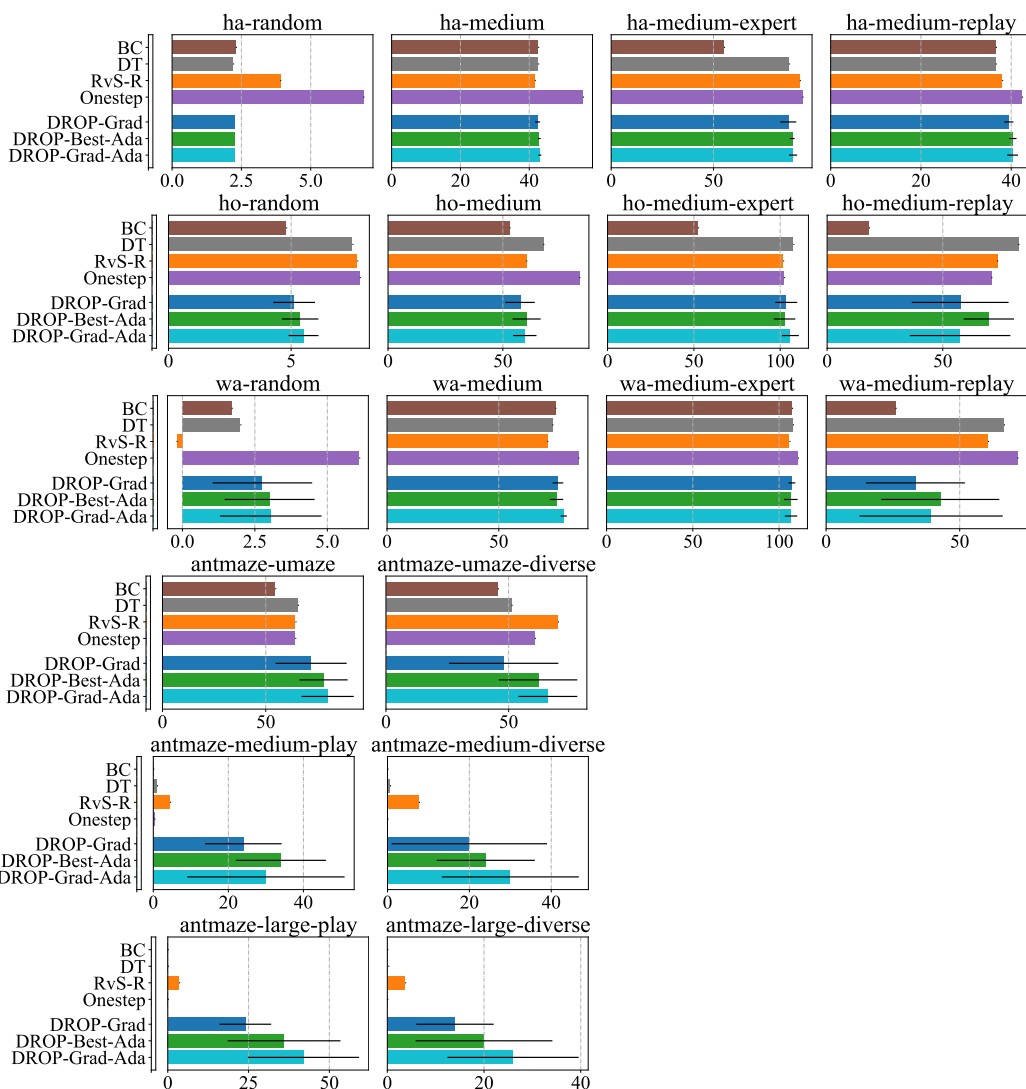

Figure 11: Comparison with non-iterative methods on D4RL (*-v2), where ha = halfcheetah, ho = hopper, and wa = walker2d. Each bar denotes the average of normalized returns. The baseline results of behavior cloning (BC), Decision Transform (DT), RvS-R, and Onestep are taken from Emmons et al. (2021). Our DROP results are computed over 5 seeds and 10 episodes for each seed.

Table 11: Normalized return of DROP-Grad, DROP-Best-Ada, and DROP-Grad-Ada accross Maze2d, AntMaze, MuJoCo-Gym, and Adroit suites.

| | | DROP-Grad | DROP-Best-Ada | DROP-Grad-Ada | DROP-Grad | DROP-Best-Ada | DROP-Grad-Ada |
|---|---|---|---|---|---|---|---|
| | | *-v0 (-v1 for Maze2d) | | | *-v2 | | |
| Maze2d | maze2d-umaze | $18.6 \pm 5.5$ | $18.7 \pm 3.6$ | $21.3 \pm 1.8$ | | | |
| | maze2d-medium | $17.0 \pm 10.6$ | $18.1 \pm 8.9$ | $24.3 \pm 5.4$ | | | |
| | maze2d-large | $26.6 \pm 15.1$ | $14.7 \pm 20.6$ | $28.8 \pm 18.8$ | | | |
| AntMaze | antmaze-umaze | $70.0 \pm 12.7$ | $72.0 \pm 18.3$ | $76.0 \pm 19.6$ | $72.0 \pm 17.2$ | $78.0 \pm 11.7$ | $80.0 \pm 12.6$ |
| | antmaze-umaze-diverse | $54.0 \pm 20.6$ | $68.0 \pm 7.5$ | $66.0 \pm 10.2$ | $48.0 \pm 22.3$ | $62.0 \pm 16.0$ | $66.0 \pm 12.0$ |
| | antmaze-medium-play | $20.0 \pm 6.3$ | $26.0 \pm 18.5$ | $30.0 \pm 6.3$ | $24.0 \pm 10.2$ | $34.0 \pm 12.0$ | $30.0 \pm 21.0$ |
| | antmaze-medium-diverse | $12.0 \pm 16.0$ | $20.0 \pm 11.0$ | $22.0 \pm 14.7$ | $20.0 \pm 19.0$ | $24.0 \pm 12.0$ | $30.0 \pm 16.7$ |
| | antmaze-large-play | $16.0 \pm 18.5$ | $20.0 \pm 21.0$ | $16.0 \pm 13.6$ | $24.0 \pm 8.0$ | $36.0 \pm 17.4$ | $42.0 \pm 17.2$ |
| | antmaze-large-diverse | $20.0 \pm 8.9$ | $22.0 \pm 24.8$ | $22.0 \pm 14.7$ | $14.0 \pm 8.0$ | $20.0 \pm 14.1$ | $26.0 \pm 13.6$ |
| MuJoCo-Gym | halfcheetah-random | $2.3 \pm 0.0$ | $2.3 \pm 0.0$ | $2.3 \pm 0.0$ | $2.3 \pm 0.0$ | $2.3 \pm 0.0$ | $2.3 \pm 0.0$ |
| | halfcheetah-medium | $36.8 \pm 0.6$ | $36.8 \pm 0.1$ | $37.1 \pm 0.4$ | $42.4 \pm 0.7$ | $42.9 \pm 0.4$ | $43.1 \pm 0.4$ |
| | halfcheetah-medium-expert | $85.6 \pm 15.1$ | $89.6 \pm 11.5$ | $96.5 \pm 7.4$ | $86.6 \pm 3.9$ | $88.5 \pm 1.2$ | $88.9 \pm 2.0$ |
| | halfcheetah-medium-replay | $32.6 \pm 8.4$ | $36.7 \pm 3.3$ | $33.6 \pm 6.8$ | $39.5 \pm 1.0$ | $40.4 \pm 0.8$ | $40.3 \pm 1.2$ |
| | hopper-random | $11.1 \pm 0.4$ | $11.1 \pm 0.4$ | $11.1 \pm 0.4$ | $5.1 \pm 0.8$ | $5.4 \pm 0.7$ | $5.5 \pm 0.6$ |
| | hopper-medium | $45.8 \pm 14.3$ | $52.2 \pm 23.1$ | $46.5 \pm 12.8$ | $57.5 \pm 6.4$ | $60.3 \pm 6.1$ | $59.5 \pm 5.1$ |
| | hopper-medium-expert | $111.2 \pm 1.8$ | $112.0 \pm 0.2$ | $112.5 \pm 0.8$ | $103.5 \pm 6.3$ | $102.5 \pm 6.2$ | $105.9 \pm 4.9$ |
| | hopper-medium-replay | $31.2 \pm 6.0$ | $32.6 \pm 3.8$ | $32.0 \pm 4.0$ | $48.0 \pm 17.7$ | $83.4 \pm 6.5$ | $87.4 \pm 2.1$ |
| | walker2d-random | $3.9 \pm 0.9$ | $4.1 \pm 0.7$ | $4.1 \pm 0.8$ | $2.8 \pm 1.7$ | $3.0 \pm 1.6$ | $3.0 \pm 1.8$ |
| | walker2d-medium | $19.2 \pm 13.4$ | $26.3 \pm 8.2$ | $24.7 \pm 9.7$ | $76.5 \pm 2.4$ | $75.8 \pm 3.0$ | $79.1 \pm 1.4$ |
| | walker2d-medium-expert | $84.4 \pm 23.2$ | $91.7 \pm 10.2$ | $91.3 \pm 5.7$ | $107.5 \pm 2.0$ | $106.8 \pm 3.9$ | $106.9 \pm 3.6$ |
| | walker2d-medium-replay | $13.1 \pm 4.8$ | $13.9 \pm 4.3$ | $16.9 \pm 7.0$ | $37.4 \pm 13.5$ | $60.9 \pm 7.4$ | $61.9 \pm 2.3$ |
| Adroit | door-cloned | $0.5 \pm 0.7$ | $2.5 \pm 2.5$ | $2.7 \pm 3.2$ | | | |
| | door-expert | $98.6 \pm 5.5$ | $102.2 \pm 2.4$ | $102.6 \pm 1.6$ | | | |
| | door-human | $3.3 \pm 4.5$ | $1.9 \pm 2.2$ | $3.0 \pm 2.6$ | | | |
| | hammer-cloned | $0.3 \pm 0.0$ | $0.3 \pm 0.0$ | $0.3 \pm 0.0$ | | | |
| | hammer-expert | $65.7 \pm 26.6$ | $73.3 \pm 15.3$ | $77.7 \pm 25.0$ | | | |
| | hammer-human | $1.1 \pm 1.6$ | $0.3 \pm 0.1$ | $2.1 \pm 3.5$ | | | |
| | pen-cloned | $76.7 \pm 15.0$ | $77.1 \pm 6.0$ | $82.4 \pm 36.6$ | | | |
| | pen-expert | $113.1 \pm 10.2$ | $118.6 \pm 13.3$ | $116.7 \pm 23.5$ | | | |
| | pen-human | $71.1 \pm 34.1$ | $85.1 \pm 13.5$ | $81.5 \pm 16.2$ | | | |
| | relocate-cloned | $0.1 \pm 0.0$ | $0.5 \pm 0.5$ | $0.2 \pm 0.2$ | | | |
| | relocate-expert | $2.5 \pm 3.3$ | $6.2 \pm 4.9$ | $5.4 \pm 3.6$ | | | |
| | relocate-human | $0.0 \pm 0.0$ | $0.0 \pm 0.0$ | $0.0 \pm 0.0$ | | | |

