# OpenReview forum: "DROP: Conservative Model-based Optimization for Offline Reinforcement Learning"
_ICLR.cc/2023/Conference — Submitted to ICLR 2023_

### Official Review · Reviewer_8eDD · 2022-10-19

**Confidence:** 4
**Correctness:** 2
**Technical Novelty And Significance:** 2
**Empirical Novelty And Significance:** 2
**Recommendation:** 6

**Clarity, Quality, Novelty And Reproducibility:**

**Clarity**

The paper needs improvements wrt structure and is very difficult to read. Terms are used before they are explained and there is no reference that they will be explained later. So you get the impression that it makes more sense to read the paper from back to front than the other way round.

**Quality**

The quality of the presentation needs to be improved drastically.

**Novelty**

The main concepts are not really new, the main ideas already exist.

**Reproducibility**

Good

**Further comments**
* The first sentence „Offline reinforcement learning describes the task of learning policy from previously collected static data.“  is wrong in my opinion. It should be „Offline reinforcement learning describes the task of learning a policy from previously collected static data.“
or
„Offline reinforcement learning describes the task of learning policies from previously collected static data.“
* Since the statement "Then, these methods directly deploy the learned policy in the online environment to test the performance." is wrong (the algorithms mentioned are offline procedures and do not test online), the authors mean something different than usual by "online environment".
* What is meant by "online environment" here?
* What is meant by "employed in testing"?
* „To enable this model-based optimization in offline RL, we are required to decompose an offline RL task into multiple sub-tasks, each of which thus corresponds to a behavior policy-return (parameters-return) pair.“ Why multiple sub-tasks? Doesn't this simply mean that supervised learning is used to learn a predictive model of the dynamic behavior and rewards of the environment from offline data (system identification (J. Sjöberg, H. Hjalmarsson, L. Ljung, Neural Networks in System Identification, 1994)) and then use this model to optimize a policy? See e.g. A.M. Schaefer, Reinforcement Learning with Recurrent Neural Networks, 2008.
* The term sub-tasks is very misleading, what is meant here are sub-sets of the data under the assumption that for each sub-set exactly one policy was active and that for different sub-sets different policies were active.
* What is meant by "unconfident embeddings"?
* The terms "low-dimensional embeddings for decomposed sub-tasks", "deployment adaptation" are used before they were explained.
* The statement "Such decomposition also comes with an additional benefit that it provides an avenue to exploit the hybrid modes in offline data D, because that D is often collected using hybrid data-generating behavior policies (Fu et al., 2020)," is, in my opinion, unacceptable. I read this as: because the benchmark datasets were created in this particular way, an algorithm is now proposed that exploits this property of the benchmark datasets. There is no practical benefit from this.




**Strength And Weaknesses:**

**Strengths**
* The algorithm may be useful in special cases that may arise in practice.

**Weaknesses**
* The presentation needs to be improved significantly wrt structure and writing.
* The paper proposes an algorithm that exploits a property of the offline data in the D4RL benchmarks. It does not discuss whether this property can be expected in practice.

**Summary Of The Paper:**

The paper deals with offline RL. It takes the position that iterated policy evaluation can be given so little trust that behavior cloning is preferable.
It makes the assumption that it is useful to decompose the data set into episodes and (assuming that different policies were active in different episodes) to create individual behavior clonig policies for individual episodes or appropriate groups of episodes.
The algorithm is evaluated on the D4RL benchmark, where exactly this assumption is realized.

**Summary Of The Review:**

The paper is very hard to read.
The paper proposes an algorithm that exploits a property of the offline data in the D4RL benchmarks. It does not discuss whether this property can be expected in practice.

---

> ### Author Response · Authors · 2022-11-16
> **Response to Reviewer 8eDD (2/2)**
>
>
>
> **Regarding the other detailed questions:**
>
> **Q.6: The first sentence "Offline reinforcement learning describes the task of learning policy from previously collected static data" (in Section Introduction) is wrong.**
>
> **A.6:** Thank you for the catch. It is a typo. *"learning policy"* should be *"learning a policy"*.
>
> **Q.7: The third sentence in Section Introduction. What is meant by "online environment"? What is meant by "employed in testing"?**
>
> **A.7:** "Online environment" denotes the testing environment. The evaluation of any offline RL algorithms requires deploying the trained policy to an interactive (online) environment and thus testing the performance. "Employed in testing" means we deploy the learned policy in a testing environment to test the performance.
>
> **Q.8: The term sub-tasks ...?**
>
> **A.8:** The term "sub-task" refer to we learn an offline policy using the corresponding (offline) sub-dataset.
>
>
> **Q.9: What is meant by "unconfident embeddings"?**
>
> **A.9:** In model-based optimization (not model-based RL), we use a score model to infer a new design input. If we directly perform gradient ascent on the design input over a learned pseudo score model $f$, $x_k + \nabla_xf(x)|_{x=x_k}$, it tends to drive the candidate design input towards out-of-distribution (OOD), invalid, and low-scoring inputs, as these are falsely and over-optimistically scored by the learned score model. And such OOD, invalid and low-scoring inputs denote the "unconfident" inputs ("embeddings" in the paper's context).
>
> Also, for a more detailed problem setting of model-based optimization, please refer to the paper COMs [4].
>
> [4] Brandon Trabucco, Aviral Kumar, Xinyang Geng, and Sergey Levine. Conservative objective models for effective offline model-based optimization
>
> **Q.10: The terms "low-dimensional embeddings for decomposed sub-tasks", and "deployment adaptation" are used before they were explained.**
>
> **A.10:** Thank you for the suggestion. We will discuss them accordingly.
>
> **Q.11: The statement "... exploit the hybrid modes in offline data ..." is, in my opinion, unacceptable. I read this as: because the benchmark datasets were created in this particular way, an algorithm is now proposed that exploits this property of the benchmark datasets. There is no practical benefit from this.**
>
> **A.11:** Thanks for raising this concern, but we respectfully disagree with your comment. We kindly point out that *the D4RL benchmark dataset itself has the characteristics of multi-modality*. For example, the \*-diverse domain dataset means the dataset contains diverse behaviors; the behavior policies for Gym-MuJoCo domain tasks are learned with SAC, which adopts stochastic policy networks and thus can demonstrate a multi-modal offline data distribution.
>
>
> Thank you for your review again. We hope we have resolved your concerns. We are always willing to answer any of your further concerns.

---

> > ### Comment · Reviewer_8eDD · 2022-11-17
> > **Quick Response**
> >
> > I welcome the detailed response from the authors. Before I address the individual points, I would like to try in advance to clear up the misunderstanding that still exists about `The paper proposes an algorithm that exploits a property of the offline data in the D4RL benchmarks. It does not discuss whether this property can be expected in practice.`
> > What I am asking for is that the authors give reasons in the paper why it makes sense to assume
> > * that different behavior policies have been used in the data, and
> > * that among these policies there is at least one that is so good that it is beneficial to copy it. And
> > * that, furthermore, the different policies were not used in an arbitrary way, but in such a way that whole trajectories can be assigned to one policy each. Which in turn also means that the number of different policies is not greater than the number of episodes. Which in turn also means that no continuously changing policy has been used.
> >
> > I expect the authors to give practical examples of this, or at least to name papers that give practical examples, and to do so in a way that the reader can recognize "practical examples are given in that paper". The reference that such datasets are included in D4RL is not sufficient.
> >
> > If the authors do not want to or cannot give practical examples, then it should be pointed out very early (e.g. in the title and/or abstract) that the paper deals with the special case that different behavior policies have been used trajectory-wise in the data.
> >
> > To avoid further confusion; I know that in practice there are examples where the assumption makes sense. But there are also examples where it does not make sense. Accordingly, the text must be designed in such a way that it is clear that this paper is not about a general approach, but that a special case is addressed. Ideally, the paper should also explain why this special case is worth addressing.

---

> > > ### Author Response · Authors · 2022-11-17
> > > **Clarification about our assumption (practical offline data is multi-modal)**
> > >
> > > Thank you for your timely response and detailed comments. We note that the reviewer summarized three assumptions. Next, we will reply to each of them.
> > >
> > > **different behavior policies have been used in the data**
> > >
> > > In practice (real-world offline tasks), offline data are often collected by *multiple* human collectors in *multiple* different environments, which, therefore, exhibits *multiple* behavioral patterns. For example, for medical treatment datasets, we can collect diagnostic data from different patients at different times; for self-driving datasets, even the same driver may exhibit different behavioral patterns in the same state. *Thus, we believe it is reasonable to assume that the practical offline dataset is multi-modal.*
> > >
> > > Besides, if we assume that offline data do not have the characteristics of multimodality (offline data is collected by a fixed behavioral policy in a deterministic environment), then we can only learn a single behavior policy (corresponding to a fixed embedding). In such cases, our DROP method will degenerate to the Onestep model.
> > >
> > > To clarify these practical assumptions, we will add additional expositions in our paper.
> > >
> > > **among these policies there is at least one that is so good that it is beneficial to copy it**
> > >
> > > We kindly point out that we do not make such an assumption. Our DROP method does not simply copy the best behavior in the offline data but tries to *find (extrapolate) a better policy beyond all behaviors in the dataset*. We refer the reviewer to Figure 3, where we show that our approach is essentially stitching together multiple behavioral policies, rather than simply choosing the best one. Besides, quantitatively, our approach also achieves a better performance than the baselines on \*-random tasks (Table 3).
> > >
> > > **The number of different policies is not greater than the number of episodes. Which in turn also means that no continuously changing policy has been used.**
> > >
> > > Our naive DROP implementation performs task decomposition according to the returns of trajectories. We agree with the reviewer that such an implementation may cause the issue mentioned by the reviewer. However, instead of simply conducting return-guided task decomposition, our DROP+CVAE implementation adopts CVAE to model the offline data and afterward takes the learned latent embedding in CVAE as the behavior embedding. We think such a CVAE-based implementation can resolve the concern raised by the reviewer.
> > >
> > > We hope that these clarifications have addressed your concerns. We are always willing to answer any of your further concerns.

---

> > > > ### Comment · Reviewer_8eDD · 2022-11-17
> > > > **Important points - good response**
> > > >
> > > > From my point of view, all these points are very important, and they should be presented in the paper in such a way that also the future readers get a clear picture of
> > > > * the practical relevance,
> > > > * the limitations, and
> > > > * the performance of the method which goes beyond the obvious.
> > > >
> > > > However, I would still ask that the authors do not write that a data set with multiple behavior policies is the usual or even only situation (as in `practical offline data is multi-modal`). Because that is too one-sided. There is a wide range of situations, including the case where a single, fixed policy generated the data, as well as the case where the assumption that a plurality of behavior policies generated the data falls short because, for example, the agent(s) that generated the data, be they humans, classical controllers, or RL policies, use more information than is available to the policy to be learned via offline RL. (Meaning that the agents that produced the data cannot be represented in the state space chosen for the offline RL approach).

---

> > > > > ### Author Response · Authors · 2022-11-17
> > > > > **Thank you for the suggestion**
> > > > >
> > > > > We thank the reviewer for the detailed and valuable comments. In our new version, we will add the practical relevance (to the assumption), limitations, and more descriptions (wrt our experimental results). Besides, we will modestly depict the scope of this paper as suggested by the reviewer.
> > > > >
> > > > > Thanks again for improving this paper and giving very constructive comments. We hope we have resolved your concerns. We are always willing to answer any of your further concerns.

---

> > > > > > ### Comment · Reviewer_8eDD · 2022-11-19
> > > > > > **Score Adjustment**
> > > > > >
> > > > > > It is still too early to finalize the recommandation, but I am increasing the score now in order to avoid too big a discrepancy between my current and still preliminary assessment and the Recommandation.

---

> > > > > > > ### Author Response · Authors · 2022-11-19
> > > > > > > **Response to Reviewer 8eDD**
> > > > > > >
> > > > > > > We'd like to thank you again for your time and efforts in providing a valuable review and carefully judging our feedback. We really enjoy the communication, and it helps us make our paper better.

---

> ### Author Response · Authors · 2022-11-16
> **Response to Reviewer 8eDD (1/2)**
>
> Thanks for your comments and questions. We very much appreciate the time you took to review our work. We reply to your points below.
>
> After our careful analysis of the reviewer's comments, *we feel that the reviewer may have misunderstood our paper*. To clarify the misunderstanding, we will first address the high-level and most important concerns, and then answer all the other detailed questions.
>
> **Regarding the high-level and most important issues:**
>
> **Q.1: “To enable this model-based optimization in offline RL...”. Doesn't this simply mean that supervised learning is used to learn a predictive model of the dynamic behavior and rewards of the environment from offline data and then use this model to optimize a policy?**
>
> **A.1:** We think there is a HUGE misunderstanding here. **We friendly emphasize that our method is based on *model-based optimization* instead of *model-based RL*.** The two terminologies have totally different meanings. We refer the reviewer to Section 2.2 and the most related work (COMs [1]) for the detail wrt offline model-based optimization. To avoid such a misunderstanding, we will add new expositions in our paper to emphasize this difference.
>
> [1] Brandon Trabucco, Aviral Kumar, Xinyang Geng, and Sergey Levine. Conservative objective models for effective offline model-based optimization
>
> **Q.2: (Summary Of The Paper) It takes the position that iterated policy evaluation can be given so little trust that behavior cloning is preferable. It makes the assumption that it is useful to decompose the data set into episodes and (assuming that different policies were active in different episodes) to create individual behavior cloning policies for individual episodes or appropriate groups of episodes.**
>
> **A.2:** We thank the reviewer for summarizing our paper, but we respectfully disagree with the "Summary Of The Paper" by the reviewer. When we try to understand the "Summary Of The Paper" comments, we feel that the reviewer seems to be only focusing on our task decomposition step (Section 3.1). We kindly point out that both the conservative model optimization (Section 3.2) and the deployment adaptation (Section 3.3) are important for our DROP implementation.
>
> Meanwhile, we do not simply believe that "behavior cloning (BC) is preferable". In Table 1, we also pointed out the potential issues of the previous BC-based method (e.g., RvS-R [2]), but our method can well overcome these issues. In experiments, we can also see that our method exhibits better performance.
>
> [2] Scott Emmons, Benjamin Eysenbach, Ilya Kostrikov, and Sergey Levine. Rvs: What is essential for offline rl via supervised learning?
>
> **Q.3: (Weaknesses) The paper proposes an algorithm that exploits a property of the offline data in the D4RL benchmarks. It does not discuss whether this property can be expected in practice.**
>
> **A.3:** If we understand the reviewer's comments correctly, the "property" emphasized by the reviewer here refers to the assumption that we assume the offline data distribution is multi-modal, and the reviewer may think this assumption is too strong. However, we kindly argue that this assumption is reasonable: in practical settings, offline datasets are often heterogeneous and are collected using different behavior policies, leading to a data distribution with multiple modes. These data-collection behavior policies may aim to accomplish tasks that are not necessarily aligned with the target task or may accomplish the same task but provide conflicting solutions [3]. Moreover, the standard D4RL dataset also exhibits the "multi-modal" property. For example, the \*-diverse domain dataset itself contains diverse behaviors; the \*-medium dataset is generated by a policy learned using SAC, which adopts stochastic policy networks and thus demonstrates a multi-modal offline data distribution.
>
> [3] Xi Chen, Ali Ghadirzadeh, Tianhe Yu, Yuan Gao, Jianhao Wang, Wenzhe Li, Bin Liang, Chelsea Finn, and Chongjie Zhang. Latent-variable advantage-weighted policy optimization for offline rl.
>
>
> **Q.4: (Novelty) The main concepts are not really new, the main ideas already exist.**
>
> **A.4:** Can the reviewer point out this problem more clearly? We note that we have received the novelty acknowledgment from both reviewers f5fq and rqxp.
>
> **Q.5: The paper needs improvements wrt structure.**
>
> **A.5:** Thank you for the suggestion. In our revised paper, we have added new descriptions wrt the overall framework in the method section and moved some important expositions (core algorithms and experimental results) from the appendix to the main paper.

---

### Official Review · Reviewer_rqxp · 2022-10-25

**Confidence:** 4
**Clarity, Quality, Novelty And Reproducibility:** 1. Clarity and quality
**Correctness:** 3
**Technical Novelty And Significance:** 3
**Empirical Novelty And Significance:** 3
**Recommendation:** 5

**Strength And Weaknesses:**

Strength:
1. It is a reasonable idea to split the offline datasets into N sub-tasks and condition both the score model and the policy model on a task embedding z. The proposed method has two advantages:
- By explicitly considering the multi-modal distribution of the offline datasets and inferring the task embedding at test time, it can improves the domain adaptation ability of the policy.
- By optimizing z (which is compact), it can ease the high-dimensional optimization difficulty and thus extend mode-based optimization (Trabucco et al., 2021) to test-time policy adaptation.
2. Good ablation study on the embedding inference approaches and the decomposition rules (better to move into the main body of the paper).

Weaknesses:
1. The writing of the paper can be largely improved.
- The structure of this paper is not very well organized. Frankly speaking, I spent a lot of time reading this paper so that I could understand Q1-Q3 and find out how they relate to each contribution of the proposed method. It might be better to describe the overall framework and the technical novelty before going to each detailed component in the method section. When I read the method section, I felt a little bit lost until I finished Sec 3.3 which introduced the last test-time adaptation module. I suggest describing the general pipeline of the model in the first paragraph of Sec 3 for a better reading experience.
- Fig 1 is less informative, which only shows the differences between iterative and non-iterative actor-critic offline RL methods, rather than illustrating the unique properties of the proposed method. Since the work of OneStep (Brandfonbrener et al., 2021) also follows the non-iterative framework, this figure does not shed a light on the novel contribution of DROP.
- The main body of the paper is not self-contained. I have to frequently flip between the main text and the appendix. In my view, at least the core algorithms and some important quantitative experimental results (such as the full comparisons with existing offline RL methods) should be placed in the main text.
- Misleading notations. In Sec 2.1, "mu" stands for the initial state distribution, while in Eq (8), mu is the distribution of the latent sub-task embedding "z". This might raise confusion. Suggest using two different notations here.
- Use ``xxx'' for the quotation marks in LaTeX, instead of ''xxx''.
2. For the experiments,
- In Table 2, the proposed model is only compared with OneStep and COMs in the AntMaze environments, which is severely inadequate to support the advantages of the proposed method.
- The model was claimed to outperform previous value regularized methods including BEAR/BCQ/CQL on 19/33 tasks (Page 8), however, in Table 10, CQL achieves a better performance on 16/33 tasks than the proposed model. These results undoubtedly weaken the effectiveness of the proposed method, especially considering that it also borrows the regularization term from CQL.
- As a further ablation study, it would be good to show the experimental results without deployment adaptation, which can better verify the effectiveness of performing outer-level optimization in the embedding space rather than the action space from the original COMs.








**Summary Of The Paper:**

This paper presents a non-iterative actor-critic approach for offline RL, which tackles the erroneous overestimation problem of the critic by combining the ideas of the non-iterative training procedure, test-time policy adaptation (both from Brandfonbrener et al., 2021), and conservative objective optimization (from Trabucco et al., 2021).

The key insight of this paper is to fit the assumed multiple modes of offline data distribution by learning a mode-specific embedding as the condition of the actor and the critic. To this end, specific training methods include:
1. Divide the offline training datasets into multiple sub-tasks based on certain decomposition rules.
2. In the offline training phase, the policy network and the embedding network are learned jointly with behavior cloning, and the score model is learned with TD-errors.
3. In the online testing phase, slightly different from optimizing the entire policy network (Trabucco et al., 2021), the authors perform gradient ascent in the embedding space to find the optimal z* that can maximize the estimated objectives of the learned score model.

Compared with previous work, the technical contributions of this paper are:
1. The idea of task decomposition;
2. The efficient policy adaptation approach at test time by performing optimization in the compact embedding space.

**Summary Of The Review:**

In general, I think this paper proposes an interesting model to tackle the out-of-distribution learning problem in offline RL tasks, and contains rich ablation studies of different task decomposition rules and embedding inference methods. I was inclined to accept this paper given its novelty, however,
- The paper lacks sufficient results to demonstrate the performance advantage over previous iterative value regularization offline RL methods, especially considering the model incorporates the CQL regularization when learning the score model. These experiments seem unconvincing to me. I strongly suggest further validation of the model by comparing its effectiveness with CQL in the revised paper.
- The overall writing should be improved. Core algorithms and important results should not be placed in the appendix part. Please amend them accordingly.

---

> ### Author Response · Authors · 2022-11-16
> **Response to Reviewer rqxp (2/2)**
>
> **Q.8: As a further ablation study, it would be good to show the experimental results without deployment adaptation, which can better verify the effectiveness of performing outer-level optimization in the embedding space rather than the action space ... .**
>
> **A.8:** Thank you for the suggestion. If we understand the reviewer's suggestion correctly, the reviewer may want us to compare our DROP ($\\arg \\max_z f(s,a,z)$) to a pessimistic version of Onestep, which performs the outer-level optimization in the action space, i.e., $\\arg \\max_a Q_{\text{{pessimistic}}}(s,a)$. In the following table, we provide the corresponding results, showing that $\\arg \\max_a Q_{\text{{pessimistic}}}(s,a)$ sightly outperforms naive Onestep implementation, while it still underperforms our DROP method.
>
>
> |                               | DROP        | One-step        |$\\arg \\max_a Q_{\text{{pessimistic}}}(s,a)$        |
> | ----------------------------- | ---------- | ---------- | ---------- |
> | antmaze-umaze             | 80    |    64.3   |   72.4   |
> | antmaze-umaze-diverse     | 66    |    60.7   |   67.5   |
> | antmaze-medium-play       | 30    |    0.3   |   5.2   |
> | antmaze-medium-diverse    | 30    |    0   |   0   |
> | antmaze-large-play        | 42    |    0   |   0   |
> | antmaze-large-diverse     | 26    |    0   |   0   |
> | walker2d-medium-replay    | 61.9    |    66.4   |   69.9   |
> | hopper-medium-replay      | 87.4    |    77.3   |   78.5   |
> | halfcheetah-medium-replay | 40.3    |    38.4   |   34.8   |
> | walker2d-medium-expert    | 106.9    |    111.8   |   108.5   |
> | hopper-medium-expert      | 105.9    |    81.4   |   99.0   |
> | halfcheetah-medium-expert | 88.9    |    77   |   83.3   |
> | total | 765.3    |    577.6   |   619.1   |
>
>
> Thank you for your review again. We hope we have resolved your concerns. We are always willing to answer any of your further concerns.

---

> ### Author Response · Authors · 2022-11-16
> **Response to Reviewer rqxp (1/2)**
>
> We would like to sincerely thank the reviewer for the detailed review and the encouraging comments:
>
> > In general, I think this paper proposes an interesting model to tackle the out-of-distribution learning problem in offline RL tasks, and contains rich ablation studies of different task decomposition rules and embedding inference methods.
>
> We will first address the high-level and most important concerns, and then answer all the other detailed questions.
>
> **Regarding the high-level and most important issues:**
>
> **Q.1: (The writing/structure of this paper.) I suggest describing the general pipeline of the model in the first paragraph of Sec 3 for a better reading experience.**
>
> **A.1:** Thank you for the suggestion. In our revised paper, we have added new descriptions wrt the overall framework in the method section and moved some important expositions (core algorithms and experimental results) from the appendix to the main paper.
>
> **Q.2: I strongly suggest further validation of the model by comparing its effectiveness with CQL in the revised paper.**
>
> **A.2:** Thank you for your suggestion. In our revised paper, we added CQL's performance and new Gym-\*-medium-\* tasks in Table 2. We can find that, compared with CQL, our naive DROP implementation shows superior performance in AntMaze-large-* and Gym-\*-medium-expert (m.-exp.) tasks, while it leads to inferior performance in AntMaze-medium-* and Gym-\*-medium-replay (m.-rep.) tasks. As an extension, we also compare CQL with our DROP+CVAE implementation (see details in Appendix C.4). We can find DROP+CVAE further improves DROP’s performance and retains superior/comparable performances in all tasks (see Gym-\*-medium-* tasks in Table 2, see Gym-\*-medium and Gym-\*-random tasks in Table 6).
>
> **Regarding the other detailed questions:**
>
> **Q.3: (The structure of this paper.)  In my view, at least the core algorithms and some important quantitative experimental results should be placed in the main text.**
>
> **A.3:** Thank you for the suggestion. We have moved some important expositions (core algorithms and experimental results) from the appendix to the main paper.
>
> **Q.4: Fig 1 is less informative, which only shows the differences between iterative and non-iterative actor-critic offline RL methods, rather than illustrating the unique properties of the proposed method.**
>
> **A.4:** We present Figure 1 out of concern that readers may not be familiar with the difference between iterative and non-iterative offline RL paradigms. At the same time, we raise three core questions from Figure 1, which we think is the core outline of the whole paper. Thank you for raising this concern again. We will revise the corresponding exposition of the paper.
>
> **Q.5: In Sec 2.1, "mu" stands for the initial state distribution, while in Eq (8), mu is the distribution of the latent sub-task embedding "z".**
>
> **A.5:** Thank you for the detailed review. We have changed the corresponding notation.
>
> **Q.6: Use ``xxx'' for the quotation marks in LaTeX, instead of ''xxx''.**
>
> **A.6:** Thank you for the suggestion. We will change the corresponding quotation mark.
>
> **Q.7: In Table 2, the proposed model is only compared with OneStep and COMs in the AntMaze environments**
>
> **A.7:** Thank you for raising this concern. We have added new experiments (Gym-\*-medium-\*) in Table 2 (we refer the reviewer to our revised paper). We can find that these new results also support the main claim in our paper.

---

> ### Author Response · Authors · 2022-12-07
> **Discussion period ends soon**
>
> Dear Reviewer rqxp,
>
> It is a kind reminder that the Reviewer-Author discussion phase is coming to an end. Following your suggestion, we believe that we have made a great effort to provide all the clarifications and experiments.
>
> If you have read our latest response, please kindly let us know. Any further questions/discussions are welcome.
>
> Thanks again for your review. Looking forward to your reply. Thank you!

---

### Official Review · Reviewer_X8xm · 2022-10-25

**Confidence:** 4
**Correctness:** 2
**Technical Novelty And Significance:** 2
**Empirical Novelty And Significance:** 3
**Recommendation:** 3

**Clarity, Quality, Novelty And Reproducibility:**

+ The approach seems to be reproducible and some practical implications are presented.

- Although the paper is well-written, some parts are a bit hard to understand. The clarity of contributions, approach, and argumentation should be improved. In addition, providing some context/background (e.g., on the terms) from the beginning would be helpful.

- The connection of paper with the related work can be clarified further, particularly to the most closely related work. In addition, a line of research on "model-based" offline RL is missing from the paper (see references mentioned below). In this area, model is referred to the dynamics model of the environment which is first estimated and then used for policy learning. Whereas, the model in this paper is interpreted differently (i.e., parametric embeddings). This connection should be clarified/discussed.

- Apart from "model", the term "offline" seems to be loosely used. The approach delays the policy inference to the testing time when the model can acquire new samples (see Fig.2 and Alg. 2). While to my knowledge, in offline learning, we don't have access to online rollouts  whatsoever. Hence, it is not clear to me how the proposed ideas contribute to the field of MB offline RL (both in terms of methodology and empirical study).

- Accordingly, from where P(.|s,a) in Alg. 2 comes from?

- Could you elaborate how splitting the data into N subsets benefits the approach? An ablation study on the effect of N and also when N=1 would be helpful.


[1] Yu et al., MOPO: Model-based Offline Policy Optimization (2020)
[2] Kidambi et al., MOReL: Model-Based Offline Reinforcement Learning (2020)
[3] Yu et al., COMBO: Conservative Offline Model-Based Policy Optimization (2021)
[4] Rigter et al., RAMBO-RL: Robust Adversarial Model-Based Offline Reinforcement Learning (2022)

**Strength And Weaknesses:**

Strengths:
+ The paper is well-written and visually appealing.
+ The idea of decoupling conservative behavioral cloning from policy inference is interesting and somewhat novel.
+ The experiments are extensive and reproducible, and the results are promising on the D4RL benchmark data.



Weaknesses:
- The clarity of the paper in terms of contributions, approach, and argumentation should be improved
- The literature is not sufficiently reviewed and the position of the paper in the domain is not clear.
- The significance of the contributions on the model-based offline RL domain is not completely clear (see comments below)

**Summary Of The Paper:**

The paper presents DROP, a non-iterative framework for policy optimization in offline RL, which separates the model training on offline data from deploying the model for testing/adaptation. The approach first splits the data into N subsets to learn distinct behavior policies, which are further used to learn a contextual behavior policy and task embeddings for a conservative model optimization to be used for policy inference. Experiments on D4RL benchmark data show that the proposed work performs better or on par compared to several competitors.

**Summary Of The Review:**

The paper is well-written and introduces a novel approach for offline RL that separates policy inference from the offline training of a conservative model. However, the position of the contributions in the domain of model-based offline RL is not clear and the paper should be improved in terms of clarity. I hence vote for a reject.

---

> ### Author Response · Authors · 2022-11-16
> **Response to Reviewer X8xm (2/2)**
>
> **Q.7: Could you elaborate how splitting the data into N subsets benefits the approach? An ablation study on the effect of N and also when N=1 would be helpful.**
>
> **A.7:** We appreciate your suggestion. To check if multiple behavior policies are important for results, we ablate the different number of behavior policies in the offline data decomposition step. In the table below, we provide the ablation results for DROP-Best. We can find that DROP with only a few behavior policies can not guarantee useful policy, especially for the Antmaze-medium and Antmaze-large tasks. Such an observation is also consistent with the results of other non-iterative methods that learn a single behavior policy (see results in Table 2).
>
> | Number  of behavior policies | 1    | 10   | 50   | 100  | 200  | 300  | 400  | 500  | 600  | 800  | 1000 |
> | ---------------------------- | ---- | ---- | ---- | ---- | ---- | ---- | ---- | ---- | ---- | ---- | ---- |
> | antmaze-umaze-v2             | 0    | 76.7 | 66.7 | 50   | 73.3 | 66.7 | 56.7 | 60   | 73.3 | 63.3 | 56.7 |
> | antmaze-umaze-diverse-v2     | 0    | 53.3 | 43.3 | 60   | 46.7 | 76.7 | 63.3 | 52.5 | 40   | 45   | 43.3 |
> | antmaze-medium-play-v2       | 0    | 0    | 10   | 36.7 | 56.7 | 36.7 | 30   | 26.7 | 36.7 | 23.3 | 30   |
> | antmaze-medium-diverse-v2    | 0    | 0    | 10   | 16.7 | 10   | 16.7 | 16.7 | 20   | 10   | 13.3 | 16.7 |
> | antmaze-large-play-v2        | 0    | 0    | 20   | 20   | 23.3 | 13.3 | 6.7  | 30   | 20   | 16.7 | 6.7  |
> | antmaze-large-diverse        | 0    | 0    | 53.3 | 40   | 13.3 | 23.3 | 20   | 16.7 | 23.3 | 10   | 3.3  |
>
> In addition, we kindly point out that we did not choose the size of subsets N manually. In our implementation, we take the number of sub-tasks N as a hyperparameter, and follow the normal practice of hyperparameter selection to choose the size of N. We refer the reviewer to our implementation details section in Appendix D.
>
>
>
> Thank you for your review again. We hope we have resolved your concerns. We are always willing to answer any of your further concerns.

---

> ### Author Response · Authors · 2022-11-16
> **Response to Reviewer X8xm (1/2)**
>
> We would like to thank the reviewer for your time and encouraging comments:
>
> > The paper is well-written. The idea of decoupling conservative behavioral cloning from policy inference is interesting. The experiments are extensive and reproducible.
>
> We will first address the high-level and most important concerns, and then answer all the other detailed questions.
>
> **Regarding the high-level and most important issues:**
>
> **Q.1: The position of the contributions in the domain of model-based offline RL is not clear and the paper should be improved in terms of clarity. I hence vote for a reject.**
>
> **A.1:** We notice that the main concern raised by Reviewer X8xm is that "the position of the contributions in the domain of model-based offline RL is not clear". **We friendly point out that our method is based on *model-based optimization* instead of *model-based RL*.** The two terminologies have totally different meanings. To avoid such a misunderstanding, we will use the abbreviation “MBO” to replace “model-based optimization” in our paper, and add new expositions to emphasize the difference.
>
> **Q.2: The approach delays the policy inference to the testing time when the model can acquire new samples (see Fig.2 and Alg. 2). While to my knowledge, in offline learning, we don't have access to online rollouts whatsoever.**
>
> **A.2:** Thank you for raising this concern. We think there is a HUGE misunderstanding here: It is true that, in offline learning, we do not have access to online rollouts. However, the terminology "offline" here emphasizes that *the policy training phase* is offline and can not access online rollouts. *In the testing phase (deployment-time phase)*, we must deploy the learned policy in an (online) environment. Thus, at testing-time, we can access the agent's state $s_t$. Typical offline methods directly sample actions from the policy network (e.g., $a_t\sim\pi(a_t|s_t)$). However, we conduct two steps to sample actions (Lines 3 and 4 in Algorithm 2). To make it clear, we have modified Figures 1 and 2 to further emphasize the differences with online fine-tuning (please refer to our revised paper).
>
>
> *Please allow us to emphasize once again that our Algorithm 2 refers to the offline testing phase, not the offline training phase.*
>
>
> **Regarding the other detailed questions:**
>
> **Q.3: The connection of paper with the related work can be clarified further, particularly to the most closely related work.**
>
> **A.3:** As noted by Reviewers f5fq and rqxp, the most closely related work is COMs [1] (, Onestep and RvS). We refer the reviewer to Section 3.4 for the connection. We speculate that the reviewer may want us to clarify the differences (or connections) between *model-based rl* and our paper (that is actually based on *model-based optimization*). In the paper, we will add new descriptions to clarify this difference.
>
> [1] Brandon Trabucco, Aviral Kumar, Xinyang Geng, and Sergey Levine. Conservative objective models for effective offline model-based optimization
>
> **Q.4: A line of research on "model-based" offline RL is missing from the paper.**
>
> **A.4:** we refer the reviewer to our related work section and we have added the lost related work (RAMBO-RL) suggested by the reviewer.
>
> **Q.5:This connection (terminology "model" in our method and model-based offline RL) should be clarified/discussed.**
>
> **A.5:** Thank you for your kind suggestion. We have clarified this difference in our revised paper.
>
> **Q.6: from where P(.|s,a) in Alg. 2 comes from?**
>
> **A.6:** $P(.|s,a)$ denotes the rollout step in RL *testing-time*, where we run the policy in an accessible environment to test the performance.

---

> ### Author Response · Authors · 2022-12-07
> **Discussion period ends soon**
>
> Dear Reviewer X8xm,
>
> It is a kind reminder that the Reviewer-Author discussion phase is coming to an end. Following your suggestion, we believe that we have made a great effort to provide all the clarifications.
>
> If you have read our latest response, please kindly let us know. Any further questions/discussions are welcome.
>
> Thanks again for your review. Looking forward to your reply. Thank you!

---

### Official Review · Reviewer_f5fq · 2022-11-01

**Confidence:** 4
**Correctness:** 3
**Technical Novelty And Significance:** 3
**Empirical Novelty And Significance:** 3
**Recommendation:** 6

**Clarity, Quality, Novelty And Reproducibility:**

The quality of the paper is high. The method is novel. The reviewer cannot assess the reproducibility of the experiments.

**Strength And Weaknesses:**

This is a very novel paper that provides a fascinating new insight into Offline RL based on the COMs [1] framework.
The extension on top of COMs is simple, but the observations made in the paper (about how we learn $Q$ and how we use $Q$ to find policy) are nontrivial and fascinating.

Technical Questions:
1. For deployment-time adaptation (Figure 2 Right side, Algorithm 2), what exactly is adapting? It doesn't seem like the authors are performing parameter updates in $\beta(a|s, z^*)$ (contextual policy) or score function $f(s, a, z)$. So what is the adaptation here? When I first read it, I thought the authors were doing online learning (aka online fine-tuning), but it doesn't seem to be the case.
2. At test time, performing gradient ascent to get $z^*$ seems computationally expensive. Can authors comment on how computationally efficient this algorithm is (compared to other algorithms like CQL)?
3. How does the introduction of context $z$ help solve the OOD penalization problem? In another way, why searching $\arg\max_a Q(s, a)$ for policy is bad, while searching $z^* = \arg\max_z f(s, a, z)$ then decode as $\beta(a|s, z^*)$ is better? Because the process seems similar, the authors also trained $f$ to output a lower score when $z$ is OOD. I don't know why this is better than the pessimistic approach on the $(s, a)$ space.
4. For Table 1, would all the iterative Offline RL Algorithms (i.e., BCQ, CQL, COMBO, AWAC) satisfy all Q1, Q2, Q3? So DROP essentially is just an improvement over supervised learning for RL methods? Although, I should point out that DROP is doing TD-error learning in Equation (8), so I'm not sure why the authors think it's comparable to RvS, Decision Transformer, etc.

Strengths:
1. The experiments are very comprehensive.
2. The authors demonstrated expert-level knowledge, summarizing and abstracting prior work into a new framework that could potentially provide insight into the problem domain.

[1]: Conservative Objective Models for Effective Offline Model-Based Optimization

**Summary Of The Paper:**

This paper tries to summarize current offline RL algorithms into a bi-level optimization problem and offers a new algorithm in the non-iterative camp (that is sufficiently different from previous non-iterative work such as RvS).

**Summary Of The Review:**

I'm currently giving this paper a 6 because I think it's well-written and offers interesting insights. I also have a series of questions regarding the technical and conceptual parts of the paper. Based on the answer, I will re-assess my score for this paper.

---

> ### Author Response · Authors · 2022-11-16
> **Response to Reviewer f5fq (3/3)**
>
> **Q.4.1: For Table 1, would all the iterative offline RL Algorithms (i.e., BCQ, CQL, COMBO, AWAC) satisfy all Q1, Q2, Q3? So DROP essentially is just an improvement over supervised learning for RL methods?**
>
> **A.4.1:** Thank you for the insightful comments. We think most (iterative) offline RL baselines do not satisfy Q3 (i.e., the deployment adaptation), where many prior works (i.e., BCQ, CQL, IQL, COMBO, AWAC) assume the learned policy is fixed when interacting with the test environment. Thus, DROP is more than an improvement over the non-iterative offline RL paradigm (supervised learning), but a general guidance over the offline RL problems, suggesting that we can adaptively update the rollout policy in deployment-time in both iterative and non-iterative offline RL formulations.
>
> To the best of our knowledge, the only two offline RL counterparts that emphasize deployment adaptation are APE-V [3] and Diffuser [4]. Thus, we provide comparison results in the following table, where we can find our DROP+CVAE achieves similar performance to APE-V and outperforms Diffuser in all.
>
> |                               | DROP+CVAE        | APE-v        | Diffuser    |
> | ----------------------------- | ---------- | ---------- | ---------- |
> | walker2d-medium-replay    | 83.5    |    82.9   |     61.2    |
> | hopper-medium-replay      | 96.3    |    98.5   |     96.8    |
> | halfcheetah-medium-replay | 50.9    |    64.6   |     42.2    |
> | walker2d-medium-expert    | 109.3    |    110.0   |   108.4   |
> | hopper-medium-expert      | 107.2    |    105.7   |   107.2   |
> | halfcheetah-medium-expert | 102.2    |    101.1   |   79.8    |
>
> [3] Dibya Ghosh, Anurag Ajay, Pulkit Agrawal, and Sergey Levine. Offline rl policies should be trained to be adaptive.
>
> [4] Michael Janner, Yilun Du, Joshua B. Tenenbaum, and Sergey Levine. Planning with Diffusion for Flexible Behavior Synthesis.
>
> **Q.4.2: I should point out that DROP is doing TD-error learning in Equation 8, so I'm not sure why the authors think it's comparable to RvS, Decision Transformer, etc.**
>
> **A.4.2:** The main reason we compare DROP and RvS (Decision Transformer) is that both methods belong to the non-iterative branch of offline RL. We agree with the reviewer that DROP is doing TD-error learning in Equation 8. Thus, for the sake of completeness, we have moved the comparison results with CQL (iterative baselines that conducts TD-error learning) into our main paper.
>
>
> Thank you for your review again. We hope we have resolved your concerns. We are always willing to answer any of your further concerns.

---

> ### Author Response · Authors · 2022-11-16
> **Response to Reviewer f5fq (2/3)**
>
> **Q3: I don't know why this ($\\arg \\max_z f(s,a,z)$) is better than the pessimistic approach on the space ($\\arg \\max_a Q(s,a)$).**
>
> **A3:** Thank you, a very interesting question. We think introducing the additional contextual variable (sub-task embedding) $z$ can benefit from the following three aspects:
>
> + Inference over the embedding $z$ ($\\arg \\max_z f(s,a,z)$) can provide high-level abstract actions. Intuitively, deployment-time adaptation enables us to stitch some abstract "skills", which contribute to learning under the long horizon and spare reward offline tasks.
> + Introducing the additional $z$ provides an avenue to exploit the hybrid modes in offline data, considering that offline data is often collected by hybrid (or non-deterministic) data-generating behavior policies.
> + Compared to the pessimistic approach on the $(s,a)$ space, our DROP method enjoys one additional conservation: we sample actions from the latent policy $\beta(a_t|s_t,z^*_t)$, which provides a natural constraint to stay within the support of the dataset (action space) [2].
>
> Next, we also experimentally compare the performance between our DROP ($\\arg \\max_z f(s,a,z)$), Onestep ($\\arg \\max_a Q(s,a)$), and the mentioned baseline ($\\arg \\max_a Q_{\text{{pessimistic}}}(s,a)$) by the reviewer. We can find $\\arg \\max_a Q_{\text{{pessimistic}}}(s,a)$ sightly outperforms naive Onestep implementation, while it still underperforms our DROP method.
>
>
> |                               | DROP        | One-step        |$\\arg \\max_a Q_{\text{{pessimistic}}}(s,a)$        |
> | ----------------------------- | ---------- | ---------- | ---------- |
> | antmaze-umaze             | 80    |    64.3   |   72.4   |
> | antmaze-umaze-diverse     | 66    |    60.7   |   67.5   |
> | antmaze-medium-play       | 30    |    0.3   |   5.2   |
> | antmaze-medium-diverse    | 30    |    0   |   0   |
> | antmaze-large-play        | 42    |    0   |   0   |
> | antmaze-large-diverse     | 26    |    0   |   0   |
> | walker2d-medium-replay    | 61.9    |    66.4   |   69.9   |
> | hopper-medium-replay      | 87.4    |    77.3   |   78.5   |
> | halfcheetah-medium-replay | 40.3    |    38.4   |   34.8   |
> | walker2d-medium-expert    | 106.9    |    111.8   |   108.5   |
> | hopper-medium-expert      | 105.9    |    81.4   |   99.0   |
> | halfcheetah-medium-expert | 88.9    |    77   |   83.3   |
> | total | 765.3    |    577.6   |   619.1   |
>
>
> [2] Wenxuan Zhou, Sujay Bajracharya, and David Held. Plas: Latent action space for offline reinforcement learning.

---

> ### Author Response · Authors · 2022-11-16
> **Response to Reviewer f5fq (1/3)**
>
> We would like to sincerely thank the reviewer for the detailed review and the encouraging comments:
>
> > This is a very novel paper that provides a fascinating new insight into offline RL. The experiments are very comprehensive.
>
> Please find our responses to specific comments and questions below.
>
>
> **Q1: For deployment-time adaptation (Figure 2 Right side, Algorithm 2), what exactly is adapting?**
>
> **A1:** For the deployment-time (test-time) adaptation, at state $s_t$, we try to adaptively *update the contextual variable (sub-task embedding)* $z^*_t$. In other words, we try to find the best contextual variable $z^*_t$ (Line 3 in Algorithm 2), and sample actions from the conditioned policy $\beta(a_t|s_t,z^*_t)$ (Line 4 in Algorithm 2).
>
> We note that the reviewer also mentioned "online fine-tuning". We kindly emphasize that "online fine-tuning" either requires future reward signals or needs full rollout trajectories *before the policy adaptation*. However, in our deployment adaptation (aka test-time adaptation), we can conduct the adaptation at any state and do not use any future (rollout) signals to fine-tune the policy (embedding $z_t$).
>
> To make it clear, we have modified Figures 1 and 2 to further point out the difference with online fine-tuning.
>
>
> **Q2: Can authors comment on how computationally efficient this algorithm is (compared to other algorithms like CQL)?**
>
> **A2:** We thank the reviewer for raising this concern. In our implementation (deployment adaptation), we conduct gradient-ascent (to find better embedding z*) over every 50 timesteps rather than over every step. We empirically select time intervals from {1, 10, 20, 50, 100, 150, 200}. As shown in the following table, we provide different inference intervals and their corresponding performances. We find that 50 (steps per inference) brings less computational burden while providing stable performance.
>
> | Method | Time intervals | Average inference time at each state | Average performance in Antmaze-*-v2 tasks |
> | ---------- | ---------- | ---------- | ---------- |
> | Onestep|      |    0.00219 (s)   |      |
> | COMs   |      |    0.00074 (s)   |      |
> | CQL    |      |    0.00062 (s)   |      |
> | Diffuser [1]    |      |    0.90615 (s)   |      |
> | Diffuser (warm-start)    |      |    0.09241 (s)   |      |
> | DROP    | 1    |    0.11146 (s)   |   43.9   |
> | DROP    | 10    |    0.01252 (s)   |   46.3   |
> | DROP    | 20    |    0.00642 (s)   |   45.9   |
> | DROP    | **50**    |    **0.00425** (s)   |   **45.6**   |
> | DROP    | 100    |    0.00298 (s)   |   39.6   |
> | DROP    | 150    |    0.00149 (s)   |   36.8   |
> | DROP    | 200    |    0.00094 (s)  |   36.1   |
>
>
> Note: the above results were tested on GeForce RTX 2080 Ti.
>
> [1] Michael Janner, Yilun Du, Joshua B. Tenenbaum, and Sergey Levine. Planning with Diffusion for Flexible Behavior Synthesis.

---

> ### Author Response · Authors · 2022-12-07
> **Discussion period ends soon**
>
> Dear Reviewer f5fq,
>
> It is a kind reminder that the Reviewer-Author discussion phase is coming to an end. Following your suggestion, we believe that we have made a great effort to provide all the clarifications and experiments.
>
> Thanks again for your review. We hope we have resolved your concerns. We are always willing to answer any of your further concerns. Thank you!

---

### Comment · Reviewer_8eDD · 2022-11-17
**The Acronym DROP**

There already exists a paper in the offline RL context that introduces the acronym "DROP".
>Jian Shen et al.,, Model-based Offline Policy Optimization with Distribution Correcting Regularization, 2021\
>"... in this paper we present density ratio regularized offline policy learning (DROP) ..."

I would therefore suggest that the authors use a different acronym.

---

> ### Author Response · Authors · 2022-11-17
> **Response to Reviewer 8eDD**
>
> Thank you for improving our paper. We notice that ICLR23 also allows authors to change the title of the submission. We will change this acronym in our camera-ready version.

---

### Author Response · Authors · 2022-11-27
**Summary of Revision**

We would like to thank the reviewers for their detailed and insightful comments. In this paper, we raise three core questions for the non-iterative bi-level offline RL optimization, *which are partially addressed by prior offline RL counterparts*. Thus, we formulate offline RL as a model-based optimization (MBO) problem, *which sufficiently answers the raised questions (e.g. deployment-time adaptation) and benefits most from the non-iterative framework (strong empirical performance and ablation studies)*.

We have made every effort to address all the reviewers' concerns and responded to the individual reviews below. We have also updated the paper with several modifications to address reviewer suggestions and concerns. Summary of updates:

+ We changed Figures 1 and 2, emphasizing the difference between our deployment adaptation and online fine-tuning;
+ We enriched the introduction (Section 1) with discussion w.r.t. prior non-iterative offline counterparts and model-based RL;
+ We added the notations A1, A2 and A3 to indicate our answers to the corresponding questions (Q1, Q2 and Q3) in the main paper;
+ We re-organized the structure of our paper, adding new descriptions w.r.t. the framework (Section 3);
+ We moved the core (training/testing) algorithms from the appendix to the main paper;
+ We moved CQL results from the appendix to Table 2;
+ We added new experimental tasks (D4RL Gym-\*-medium-\* domain) in Table 2.

All updates are highlighted in red.

---

### Decision · Program_Chairs · 2023-01-20

**Decision:**

Reject

**Justification For Why Not Higher Score:**

The presentation can be improved in various places, especially for non-expert readers.  The technical novelty is somewhat limited.  Some assumptions are made on the dataset which holds for D4RL, but more investigations are needed for its generality.

**Justification For Why Not Lower Score:**

N/A

**Metareview: Summary, Strengths And Weaknesses:**

This paper proposes a non-iterative bi-level learning method for offline RL, where the inner and outer levels correspond to training and testing, respectively.  The inner level learns a score model after partitioning the offline data, and a behavior embedding is learned with a conservative regularization.  Adaptive inference can be achieved across states in testing.  Experiments show DROP achieves similar or better performance than existing methods.

The paper addresses an important problem, and the experimental results appear encouraging.  The presentation can be improved in various places, especially for non-expert readers.  The technical novelty is somewhat limited.  Some assumptions are made on the dataset which holds for D4RL, but more investigations are needed for its generality.